# Wetspun Polymeric Fibrous Systems as Potential Scaffolds for Tendon and Ligament Repair, Healing and Regeneration

**DOI:** 10.3390/pharmaceutics14112526

**Published:** 2022-11-19

**Authors:** Joana Rocha, Joana C. Araújo, Raul Fangueiro, Diana P. Ferreira

**Affiliations:** Centre for Textile Science and Technology (2C2T), University of Minho, 4800 Guimarães, Portugal

**Keywords:** biodegradable polymers, ligament, scaffolds, tendon, wetspinning

## Abstract

Tendon and ligament traumatic injuries are among the most common diagnosed musculoskeletal problems. Such injuries limit joint mobility, reduce musculoskeletal performance, and most importantly, lower people’s comfort. Currently, there are various treatments that are used to treat this type of injury, from surgical to conservative treatments. However, they’re not entirely effective, as reinjures are frequent and, in some cases, fail to re-establish the lost functionality. Tissue engineering (TE) approaches aim to overcome these disadvantages by stimulating the regeneration and formation of artificial structures that resemble the original tissue. Fabrication and design of artificial fibrous scaffolds with tailored mechanical properties are crucial for restoring the mechanical function of the tissues. Recently, polymeric nanofibers produced by wetspinning have been largely investigated to mimic, repair, and replace the damaged tissue. Wetspun fibrous structures are extensively used due to their exceptional properties, such as the ability to mimic the native tissue, their biodegradability and biocompatibility, and good mechanical properties. In this review, the tendon and ligament structure and biomechanics are presented. Then, promising wetspun multifunctional fibrous structures based on biopolymers, more specifically polyhydroxyalkanoates (PHA), polycaprolactone (PCL), and polyethylenes, will be discussed, as well as reinforcing agents such as cellulose nanocrystals (CNC), nanoparticles, and growth factors.

## 1. Introduction

Tendons and ligaments are fibrous connective tissues, with the ability to transmit loads in between muscle and bone, and from one bone to another. These structures are designed to support and carry loads and to guarantee sufficient biomechanical performance [1,2]. Nowadays, tendon and ligament injuries are a global health problem that annually affect millions of people, involving a great clinical burden on health systems that have to face high costs associated with rehabilitations, operations, and infiltrations, among others [2,3]. Just in the United States, every year, approximately 300,000 surgical repairs of shoulders, feet, and ankles, and around 350,000 anterior cruciate ligament (ACL) reconstructive surgeries are performed, costing around $30 billion [4,5]. Such injuries limit joint mobility, reduce musculoskeletal performance, and decrease drastically human comfort and wellbeing [1]. Due to the increasing life expectancy of our society and the overall rise in engagement in sports, there is a growing prevalence in the number of people who will suffer from these type of injuries [2,3,6]. In addition, the natural healing process of the tendon and ligament is very complex, slow, and not fully understood [2,7]. Currently, the therapies that are mostly used to treat this type of injury range from surgical to conservative treatments, or even treatments using drugs, the infiltration of cells or growth factors [8,9,10,11,12,13,14,15,16,17,18,19]. Nevertheless, these therapies are not entirely effective, as reinjures are very frequent and, in some cases, fail to re-establish the lost functionality [1,2].

In recent years, great scientific advances have occurred in fields such as materials engineering, biochemistry, and physicochemistry that have allowed the development of another very promising type of therapy known as tissue engineering (TE) [20]. The concept of TE was introduced by Robert Langer and Joseph Vacanti at the end of 1993, describing the fundamental elements of tissue engineered systems composed of “a structural scaffold, a cell source, biological modulators, and mechanical stimuli.” [21]. Tissue engineering approaches aim to overcome the disadvantages of the current treatments by stimulating the regeneration and formation of neotissues in order to obtain structures that resemble the original tissues [1].

That said, advanced fibrous structures are extensively used in these applications due to their exceptional properties, such as the ability to mimic the native tissue, biocompatibility and biodegradability, easy functionalization, high tensile strength, and high resistance, among others [1,2]. 

Over the years, several fiber-based fabricating techniques and polymeric materials have been proposed for the development of TE scaffolds [1,22]. Electrospun fiber meshes have drawn attention, due to their ability to resemble the highly porous nanosized structure of native extracellular matrices. However, the high fiber packing density combined with small pore size limit cell infiltration into these meshes. Studies have shown that three-dimensional (3D) wetspun structures, composed of an assembly of randomly oriented fibers with diameters in the range of hundreds of microns, provided good in vitro cell adhesion and proliferation with good penetration of cells into the inner part of the scaffold [2,22].

Wetspinning is a controllable and simple manufacturing method that allows acquisition of micron-sized fibers. Its application in tendon and ligament tissue engineering has been recently explored, even though it is the oldest process within the manufacturing processes known as spinning, including wet-, dry-, melt-, gel-, dry-jet wet- and electro-spinning [2]. Wetspinning does not use high temperatures or voltages, so it gives the possibility of using highly biocompatible polymers based on extracellular matrixes (ECM) [23]. Another important advantage of this method is that it allows adjusting or controlling the fiber diameter, the pore size, and the porosity [24]. In the presence of mild conditions, these controllable fiber characteristics make this technique suitable for cell incorporation, as cell adhesion and proliferation are facilitated. However, the main advantage of this technique for tendon and ligament tissue engineering is the obtention of fibrillar structures [2]. 

A wide range of materials can be processed using the wetspinning method, including natural derived polymers (chitosan, collagen, gelatin, etc.) as well as native ECM components [1,25]. A family of native polyesters, polyhydroxyalkanoates (PHAs), recently recognized as natural biodegradable biomaterials, are being widely investigated for tissue engineering applications due to their bacterial source, biocompatibility, non-citotoxicity, and easy degradability [25,26]. However, the application of PHAs is restricted due to their weak mechanical and thermal characteristics, slow degradation rate, uncontrollable decomposition in clinical usages, and some other limitations [26]. To improve these properties, several reinforcing agents can be used. Cellulose nanocrystals (CNC) have been extensively used as a reinforcement material without compromising biological performance. CNC exhibit excellent properties, including thermal stability, high mechanical strength, low cost, natural abundance, inherent biocompatibility, and biodegradability [26,27,28]. On the other hand, in TE the microfibrous structures could also be functionalized using nanoparticles to give them new physical and chemical properties (increasing the strength and endurance or antimicrobial, anti-adhesive, and anti-inflammatory properties) [29,30,31,32,33,34]. For all these reasons, the use of microfibrous structures functionalized with reinforcing agents for tendon and ligament repair, healing or regeneration presents an interesting route.

## 2. Tendon and Ligaments

### 2.1. Structure, Composition, and Biomechanics

Tendons and ligaments are dense, highly organized, fibrous connective tissues, whose main function is to connect and transmit loads from muscles to bones, and from one bone to another. These structures are designed to carry loads and to guarantee sufficient biomechanical performance; bearing and reinforcing the joints as well as preventing bone luxation, which implies that they suffer great tensile strengths and high compressive forces [1,2,35]. Tendon and ligament structures are typically avascular and are mainly composed of ECM that consists mostly of collagen type I and small amounts of other collagens (types III, V, and X). These collagens represent 75% of the ECM. The proteoglycans (20%) present in tendons and ligaments include decorin, fibromodulin, biglycan, and lumican, whose main function is to organize and lubricate collagen fiber bundles. The ECM is also composed of scleraxis, tenascin C, elastin (1%), and water. The cellular component of the tissues is represented by fibroblast-like cells, termed tenocytes, or ligament fibroblasts, and by small subsets of progenitor cells [1,4,36].

The tendon and ligament structure shows a highly hierarchical organization in several distinct layers. The different organization levels are composed of collagen molecules, collagen fibrils, collagen fibril bundles, collagen fibril fascicles, proteoglycans, and elastin (Figure 1a,b) [1,37]. Their biomechanical behavior is influenced by the tissue components and the highly organized structure. For example, the elasticity of tendons and ligaments is due to the large amount of type I collagen, and the sharp ends of the fibrils provide tensile strength [1,2,36].

The mechanical properties of tendons and ligaments are still extensively studied. James et al. studied the mechanical response of tendons, showing a triphasic behavior of the tissue when strain is applied (Figure 2a) [38,39]. The first region (toe region or nonlinear) defines the behavior of tendons at deformations of up to 2% of strain, and it exhibits low values of stress. At this point, the force applied to the tissue is transferred to the collagen fibrils, inducing their contraction and rearrangement. As deformation increases, the tendon reaches the linear region (up to 4% of strain). This second region is characterized by a linear increase in stress and strain, due to the straightening and stretching of collagen molecules. Moreover, the tendons show elastic and reversible behavior. If deformation continues to increase, tendons reach the third phase (yield region), displaying a flexion of the curve (microscopic failure). In this region, the stress values slightly diminish until the defibrillation of collagen fibers occurs, and can even result in a macroscopic tissue rupture [1,2,38,39]. The ligament structure showed similar behavior when subjected to tensile load (Figure 2b) [1]. A normal and healthy tendon or ligament works between 7% and 40% failure load, depending on the type. Therefore, prosthetic implants for tendon and ligament replacement or reinforcement are intended to match or exceed the mechanical properties of the original tissue. Unfortunately, some mechanical parameters obtained for existing devices such as strength, stiffness, and elongation still do not correspond to the physiological values [1,4].

### 2.2. Treatment Strategies

The natural healing process of the tendon and ligament is very complex, slow, and not fully understood, usually resulting in weaker fibrous tissue that does not regain the strength and functionality of the unharmed tissue [2,7]. Different treatments have been suggested to address this issue, including: conservative treatments [8,9], surgical treatments (suture or grafts) [8,14,15,16], non-steroidal anti-inflammatory drugs (NSAIDs) [17,18], cell infiltration [19], growth factor infiltration [10,11], and treatments using gene therapy [12,13]. A summary of the advantages and disadvantages of each treatment can be observed in Table 1. The most commonly used treatments are further discussed below.

Due to the frequent failure of non-surgical methods, surgical intervention continues to be the preferred form of treatment [55], particularly when surgery is combined with early mobility of the damaged tendon or ligament. The methods used in this sort of treatment range from the removal of harmed tissue and suturing between the ends that have not been harmed to the use of grafts. [2]. Allografts and autografts have been found to possess good mechanical strength, as well as chemical composition and correct architectural structure, to promote cell proliferation and to stimulate new tissue growth. The main problems with autografts are the limited graft availability and donor site morbidity. The use of cadaver allografts eliminates the issue of donor site morbidity but it carries a high probability of bacterial infection, late biological incorporation, transmission of blood-borne diseases, and the need for immunosuppressive drugs to prevent tissue rejection [1,35,56]. Despite clinical advances, the failure of the grafts to regenerate the tissue is also a limitation and has higher incidence in people with tendon overstretch (strain) or tear—partially or completely—patients with severe muscle atrophies and fatty infiltrations, elderly patients, and those with systemic diseases [57]. For all these reasons, there is an urgent need for the development of new and fully effective solutions. Many health professionals and researchers are trying to develop treatments that can restore the normal structure and functionality of tendons post-injury. In this sense, TE has recently provided new hope for efficacious and complete treatments for tendon and ligament injuries [1,2].

There are numerous benefits that TE has over existing clinical therapies. It is the only method that offers a structure, typically referred to as a scaffold, that aids in maintaining the biomechanical capabilities of the tendon and ligament. This scaffold additionally acts as a skeleton for the regeneration of new tissue. It enables the incorporation of the same genes, growth factors, and cells employed in the aforementioned therapies; however, in the case of TE, this incorporation would be more regulated and targeted. In this way, diffusion issues to undesirable regions far from the damaged area would be prevented in this approach. In addition, when required, the diffusion rate of these factors can be controlled. Another advantage of this approach is that, mainly by adjusting the chosen materials, how long the scaffold will remain in the body before it degrades can be controlled. In conclusion, TE allows greater control of tendon and ligament regeneration, thereby increasing the effectiveness of the treatment [2,25]. For all these reasons, the fabrication and implantation of fibrous scaffolds derived from biological sources or synthetic scaffolds composed of biodegradable polymers are drawing noticeable attention for the injury care of tendons and ligaments [1,25].

Most polymeric materials, due to their variety, versatility, and intrinsic properties such as biocompatibility, biodegradability, and non-toxicity, have been widely used for the development of innovative tendon and ligament repair systems. Indeed, there are numerous studies that report the development of systems based on natural and synthetic polymers, as well as a mixture of both, for the production of fiber-based systems that demonstrate the advantages of using these materials in this type of application [1,58].

Recently, due to their promising properties, the development of fibers by wetspinning has been widely explored for tendon and ligament treatment. In fact, these structures are recognized to be suitable for replacing anisotropic tissues and to promote their healing. The specific structure enables them to mimic the collagen organization, to guarantee mechanical support, and to infiltrate tissue during the regeneration process. Indeed, the wetspinning technique allows the fabrication of 3D fibrous structures (scaffolds) that can mimic the ECM features [1]. In addition to mimicking the collagen organization, wetspun fibers allow the incorporation of active compounds that promote an improvement and greater effectiveness of the healing process [1]. Thus, wetspun fibers show excellent potential to be used in this type of application.

## 3. Wetspinning

Wetspinning is a simple manufacturing technique that allows the obtention of fibers from a solution using a wide range of materials, including natural-based and biodegradable polymers as well as native ECM components [1,2,23]. The versatility of this method in terms of material selection and loading strategy (e.g., incorporation in the polymer matrix or encapsulation in a core-shell structure) has enabled functionalization of polymeric fibers with a wide range of bioactive compounds, including antibiotics, growth factors, proteins, and genes [59]. As said previously, wetspinning also allows for control or adjusting of the fibers diameter, the pore size, and the porosity [24]. Indeed, different wetspinning parameters influence various aspects, such as thermodynamic and kinetics aspects, of polymer coagulation with an overall effect on the production process and the final morphology of the fibers. These, among others, include the nozzle diameter, the processing temperature, the polymer concentration and molecular weight, the solvent/non-solvent system, the solution flow rate, and the presence of additives in the solution and the coagulation bath [59]. Under mild conditions, these controllable fiber characteristics make this technique suitable for cell incorporation, as cell adhesion and proliferation are facilitated [23]. It is also possible to achieve the orientation of the fibers in the direction in which the scaffold will mainly suffer the stress just by adjusting the aforementioned parameters. This orientation is much more difficult to achieve with other technologies. The materials used in this technique also allow a very high resistance, similar to that of the original tissue [60]. In addition to all this, wetspinning has a low cost, and the fibers, once produced, are easily handled and assembled [58]. The main drawback is that it is only possible to produce micron-sized fibers, limiting its application in some cases [60].

Due to these characteristics, fibers produced by wetspinning are potentially interesting for applications in several areas, for example in the textile industry for military personal protective equipment or functional active wear, tissue engineering for the development of scaffolds, in biomedical applications, for tendon and ligament regeneration, drug delivery systems, localized implants, wound dressing systems, etc. [1,2,27,61].

Figure 3 shows a standard wetspinning set-up for laboratory research, composed by a syringe mounted on a pump and equipped with an extruding needle/nozzle immersed in a coagulation bath.

This technique is based on non-solvent induced phase separation; the polymer solution is continuously extruded through the nozzle to the solvent coagulation bath, generally using a pump. In this bath, the polymer solution turns into solid polymeric filaments as the solvent is removed. In the end, different procedures can be carried out to collect, dry, and store the fibers [2,27,59].

### Process Parameters

The wetspinning process is significantly affected by a set of processing parameters that can influence the morphology, shape, diameter, and mechanical performance of the obtained fibers. Table 2 shows this set of parameters divided into three main groups, which include solution, process, and environmental parameters. The manipulation of these parameters enables the production of homogeneous fibers with optimized properties for the final application, which is usually the desired outcome. Even though it is a simple technique, adjusting the various parameters properly is a huge challenge.

The polymer(s) molecular weight and concentration, solution viscosity, and solvent are four important variables correlated with each other that can influence the final morphology and properties of wetspun fibers. Normally, solutions containing polymers of low molecular weight associated with lower viscosity tend to produce fibers with smaller diameters when compared to solutions with polymers of high molecular weight at the same concentration. Increasing the polymer’s concentration leads to a solution with increasing viscosity [62,63]. Regarding the solvent(s), the choice of the correct one(s) depends on the polymer or mixture of polymers one intends to dissolve. Solvents with faster rate of dissolution with the ability to fully dissolve the polymer(s) are preferable to obtain fibers with the desirable properties. In addition, for wetspinning, a faster rate of coagulation is preferred because the extruded polymer needs to form a thick outside layer almost immediately upon extrusion of the solution into the coagulation bath, so as to form a fiber with some structural integrity, followed by subsequent coagulation of the core to generate the oriented filamentous structure [64]. Usually, slow solvent/non-solvent diffusion leads to uniform porous structures, since the use of a non-solvent with high coagulation power generates dense core-shell structures [65]. Therefore, the solvent/non-solvent combination must be suitable for the final application.

The needle diameter and flow rate are also very important parameters. In general, a higher needle diameter leads to the formation of fibers with a larger diameter. But with the same needle diameter, a higher speed ratio can imply larger or smaller fiber diameters depending on the solution [58,64].

## 4. Biodegradable Polymers

In this work, the type of polymers used is of particular importance; they need to be biocompatible and biodegradable polymers. With the growth in environmental awareness in recent years, the use of sustainable materials and processes is essential for the development of greener systems. That said, polymers from renewable resources have gained significant attention in recent decades due to environmental issues and the awareness of limited petroleum resources [66,67]. Biodegradable polymers, whose degradation results from the action of naturally occurring microorganisms such as bacteria, fungi, and algae are an excellent option to replace the non-biodegradable ones, and thus reduce their environmental impact [68]. This is especially important for tendon tissue engineering since degradation of polymers inside the body is essential to avoid invasive surgery.

Biodegradable polymers are usually classified into two main categories according to their origin: natural or synthetic. Natural polymers are produced by biological systems such as plants, animals or microorganisms, and include collagen, gelatin, chitosan, cellulose, alginate, silk, among others. Besides their natural origin, these polymers have several advantages such as their high abundance, biocompatibility, low toxicity, and the degradation into nontoxic and non-immunogenic components. Their promising properties make them materials with high potential for biomedical applications. On the other hand, their main drawbacks, such as their weak mechanical strength, immunogenic response, microbial contamination (i.e., endotoxin), reduced tunability, and uncontrollable degradation rate, limit their application [63,68,69]. Despite the enormous potential of these natural polymers, their complex structure makes them difficult to process by wetspinning [1]. On the other hand, synthetic biodegradable polymers can be synthesized through polymerization of monomers, and include, among others, polyesters, polyurethanes, and polyanhydrides. The inherent advantages of this type of polymer, such as their excellent mechanical properties, thermal stability, and a suitable degradation profile combined with their low-cost production makes them good candidates for medical devices, implants, and soft tissue regeneration. It is also possible to tailor the structure and properties of synthetic polymers by appropriately designing the polymers functional groups. These advantages guarantee tunable, predictable, and reproducible properties that can be varied according to the specific applications. Whereas when compared to the natural polymers, they present reduced bioactivity and lack of biocompatibility/ability to mimic body parts [1,3,65,66,68].

In order to overcome these disadvantages, researchers started to consider synthetic polymers as a reinforcement of naturally derived ones; exploring and designing different composites and multi-layered structures. In this way, it is possible to combine the biocompatibility and biological properties of natural polymers with the high mechanical strength and durability characteristics of synthetic polymers. The mechanical, chemical, and thermal properties and the diversity of application areas for this type of polymers, used individually or in combination, have been widely explored, especially for the development of polymeric fibers by the wetspinning process. In this process, the properties of the developed fibers may differ according to the polymeric materials used, and of course the selection of each polymer depends on the type of final application. At the moment, fiber-based approaches for scaffold fabrication, which mimic the fibrillar architecture of the native tissue and potentially direct cellular organization and ECM deposition, are being widely investigated and selecting the right materials is a challenge [1,68,69].

## 5. Wetspun Nanofibers as Scaffolds for Tendon and Ligament Repair, Healing and Regeneration

With tendon and ligament traumatic injuries being among the most common health problems, the need to explore new strategies to enhance the efficiency of the treatment while minimizing its side effects, and thereby improving the patient’s life quality, is crucial. The combination of wetspun fibers composed by biodegradable polymers shows great potential. Three biodegradable polymers stand out for the production of continuous fibers by wetspinning: polyhydroxyalkanoates (PHAs), polycaprolactone (PCL), and polyethylene glycol (PEG), due to their high potential for tendon and ligament tissue engineering applications. This section will explore the different research works reporting the use of biodegradable wetspun fibers for the treatment of tendon and ligament injuries.

### 5.1. Polyhydroxyalkanoates (PHAs)

PHAs are natural and nontoxic biopolyesters produced by microorganisms with remarkable properties for the next generation of environmentally friendly materials [26]. As a result of their good biocompatibility and biodegradability, they are very attractive for TE. However, the application of this polyester is restricted due to weak mechanical and thermal characteristics, slow degradation rate, uncontrollable decomposition in clinical usages, and some other limitations. Thus, in order to overcome or enhance these properties, PHAs can be varyed by modifying the surface using other polymers, enzymes and inorganic materials or even by being blended with them [26].

Poly 3-hydroxybutyrate (PHB), which was the first discovered PHA in the 1920s, and its copolymers (e.g., PHBV) are particularly attractive for TE. PHB has been used in numerous TE strategies, especially in hard tissue regeneration [70,71,72]. Doyle et al. demonstrated that materials based on PHB (copolymers or composites) produce a consistent favorable bone tissue adaptation response. PHB powder was compounded with particulate hydroxyapatite (HA) powder and then injection-molded into plaques. After extensive characterization, it was proved that there was no evidence of an undesirable chronic inflammatory response after implantation periods of up to 12 months. Moreover, bone was rapidly formed close to the material and subsequently became highly organized (Figure 4). The materials showed no conclusive evidence of extensive structural breakdown in vivo during the implantation period of the study [73]. The results also showed a layer of bone-like apatite formed within a short period on the HA/PHB composite after its immersion in an acellular simulated body fluid (SBF) at 37 °C, demonstrating high in vitro bioactivity of the composite. By varying the amount of HA in the composite, the mechanical properties and bioactivity of the HA/PHB composite could be changed [74]. The particulate HA incorporated into the PHB forms a biodegradable and bioactive composite for applications in hard tissue replacement and regeneration.

The wetspinning technique has also been investigated to overcome the aforementioned shortcomings related to thermal processing of PHA [72]. Degeratu et al. studied the cytocompatibility in vitro of poly (3-hydroxybutyrate-co-3-hydroxyvalerate) (PHBV) fibers produced by the wetspinning technique. They investigated their behavior in vitro in presence of the osteoblast-like (SaOS-2) and macrophage (J774.2) cell lines. The PHBV fibers were degraded in vitro by J774.2 cells, and erosion pits were observable. The fibers were also colonizable by SaOS-2 cells, which can spread and develop on their surface. However, despite the good cytocompatibility observed in vitro, implantation in a bone defect (drilled in rabbit femoral condyles) showed that the material was only biotolerated. There was no sign of degradation in vivo or of osteoconduction. Therefore, it was concluded that PHBV is cytocompatible but is not suitable to be used as a bone graft by itself [75].

In another study by Alagoz et al., 3D PHBV scaffolds were also produced via wetspinning. In order to promote early vascularization at the implant site, the surfaces of the structures were coated with elastin-like recombinamers (ELR) and peptide arginine-glutamic acid-aspartic acid-valine (REDV) sequences. Therefore, the scaffolds were first tested in vitro with rabbit bone marrow mesenchymal cells. The in vitro biological behavior of this type of cells revealed that attachment, proliferation, and differentiation of cells on all types of the scaffolds were not significantly different because these sequences were not recognized by them. On the other hand, when tested with human umbilical vein endothelial cells (HUVEC), the modified scaffolds attracted a higher number of cells because of the REDV sequence specific for endothelial cells. PHBV-O_2_-ELR-REDV scaffolds exhibited significantly higher cell attachment and proliferation than PHBV and PHBV-O_2_ scaffolds on days 7 and 14 [76].

Recently, many attempts have also been made to study the feasibility of PHAs for use in soft tissue engineering (e.g., cartilage engineering) [77,78,79,80,81,82]. Unfortunately, in almost every study the structures are not produced by wetspinning. Nevertheless, the combination of PHAs remarkable properties and the advantages of the wetspinning technique has come to the fore in the biomedical area, offering a unique combination with a high potential to be used as innovative structures for tendon and ligament repair, healing or regeneration. The aforementioned developed studies, where PHAs were blended for biomedical applications, are shown in Table 3.

### 5.2. Polycaprolactone (PCL)

Poly (ε-caprolactone) or polycaprolactone (PCL), is a synthetic biodegradable aliphatic polyester. It is a semicrystalline, hydrophobic polymer that has attracted considerable attention in recent years, notably in the biomedical areas. Indeed, not only because of its native biocompatibility and biodegradability but also because of its excellent mechanical properties, PCL has been extensively studied for the preparation of controlled-release drug delivery systems, nerve guides, absorbable surgical sutures, and three-dimensional (3D) scaffolds for use in tissue engineering. The versatility of PCL is due to the fact that it allows modification of its mechanical, physical, and chemical properties by blending with many other polymers easily, or copolymerization [85,86,87]. All the aforementioned properties, combined with the bioresorption and degradation rate, make it more appropriate for tissue repair and regeneration compared to other biocompatible polymers [87].

Therefore, for all the reasons mentioned above, PCL has been widely used as a biomaterial for the preparation of microfibrous scaffolds by spinning techniques. In one study developed by Zhang et al., a new wetspinning system for fabricating aligned microfiber scaffolds for oriented growth and infiltration of smooth muscle cells (SMCs) using PCL as the base material was developed. The oriented growth and infiltration of SMCs on the developed scaffolds was evaluated by scanning electron microscopy (SEM) and laser confocal microscopy. Indeed, they successfully fabricated an oriented microfiber scaffold with controllable fiber diameter and porosity where SMCs could grow in an oriented fashion along the fibers and infiltrate inside the scaffold as well as maintain SMCs phenotype. Therefore, the oriented microfiber scaffold has promising potential for regenerating blood vessels and other fibrous tissues such as tendons and ligaments and even intervertebral discs [88]. Unfortunately, the intrinsic hydrophobic nature of PCL makes it difficult to interact with biological fluids [87]. These problems can be overcome by incorporating nanoparticles, reinforcing or active agents, antibiotics or simply by mixing it with other polymers, also allowing the creation of additional properties such as antibacterial, anti-inflammatory, and antioxidant [22,28,58,83,88,89,90,91].

In another study, Neves et al. studied chitosan (CHT)/poly(3-caprolactone) (PCL) blend 3D fiber-mesh scaffolds as possible support structures for articular cartilage tissue (ACT) repair. The microfibers were successfully obtained by wetspinning of three different polymeric solutions: 100:0 (100 CHT), 75:25 (75 CHT), and 50:50 (50 CHT) wt% CHT/PCL, using a common solvent solution of 100 vol.% of formic acid, as seen in Figure 5. The 3D structures presented good integrity and stability, along with open and interconnected porosity, and a pore size range suitable for TE applications. Biological assays, SEM analysis, live-dead and metabolic activity assays showed that cells attached, proliferated, and were metabolically active over all scaffolds formulations. Cartilaginous ECM formation was also observed in all formulations. The mechanical properties increased accordingly with the PCL content. The 75 CHT formulation balanced the physical-chemical and biological properties of these new CHT/PCL blend 3D fiber-meshes for cartilage regeneration best, even though mechanical properties of the 50 CHT scaffold were better [91].

A study by Calejo et al. evaluated the development of gradient scaffolds to replicate the tendon-to-bone interface using aligned microfibers produced by wetspinning (Figure 6a). For this purpose, two formulations were considered: PCL/gelatin, to mimic tendon tissues, and PCL/gelatin incorporating nano-to-microsized hydroxyapatite (HAp) particles, to mimic bone (Figure 6b). Interestingly, the difference in composition between the two formulations was reflected in the properties of the obtained fibers. In the presence of HAp, biological studies revealed that the fibers were not only able to support cell proliferation, but also to favor cellular anisotropic alignment or to induce an osteogenic-like phenotype on human adipose-derived stem cells (hASCs). In addition, the synthesis of collagen type III by human adipose-derived stem cells could be noticed in both scaffolds. Altogether, results demonstrated the feasibility of using simple fiber processing techniques such as wetspinning to tailor the cells’ responses while having a precise control over fiber composition and topography. Moreover, the combination with advanced textile techniques allowed the development of 3D fibrous scaffolds with good properties for tendon-to-bone regeneration [58].

Therefore, fibers containing PCL and produced by wetspinning have become very promising systems to be used in the healing and repair of tissues, where induced cell contact guidance, as well as matrix deposition, are essential requirements [28]. Several developed studies are shown in Table 4, where PCL was used individually or blended with other polymers for biomedical applications.

### 5.3. Polyethylenes

PEG is the most used non-ionic hydrophilic polymer in the emerging field of polymer-based drug delivery. Since the first approved PEGylated products have already been on the market for 20 years, a vast amount of clinical experience has since been gained with this polymer [92]. PEG has outstanding properties, including high structure flexibility, good solubility in water and in organic solvents, no toxicity, and a lack of antigenicity, and consequently, no immunogenicity, all of which are essential properties for drug formulations [93,94]. The success of PEG in drug-delivery applications also led to its use in other medical fields, such as TE. Some polymers tend to have high brittleness; therefore, in order to ensure properties such as flexibility for its applications, it must be modified with additives such as plasticizers or by co-polymerization [95]. PEG has been used to form various block copolymers with PCL and PLA, and also as a plasticizer for PHAs, among others [92,93,95,96]. The use of ecofriendly plasticizers is the easiest and cheapest way to modulate the mechanical and physical and properties of biopolymers for specific applications [97].

In a study by Lu et al., bioactive and oriented collagen filaments were fabricated by a combination of wetspinning and carbodiimide-based crosslinking. The filaments consisted of an acid-dissolved collagen I solution mixed with 3.25 wt% PEG 4000 solution. The obtained filaments, which were prepared under a rotation rate, q, of 60 r/min and a perfusion rate, Q, of 0.2 mL/min, were chosen for crosslinking and further characterization. After detailed characterization, they concluded that mechanical strength of collagen filaments can be greatly improved via crosslinking. In addition, the PEG component in the spinning dope not only increased the solution content and adjusted the viscosity of the precursor, but also protected the collagen from potential structure loss during the spinning process. The results of this study also demonstrated that cells can sense and respond to the aligned wetspun collagen filaments; attaching, spreading, and proliferating along the scaffold, showing suitable biological features [98].

Interestingly, polyethylene oxide (PEO) is another name for PEG. Typically, ethylene oxide macromolecules with molecular weights of less than 20.000 g/mol are called PEG, while those having weights above 20.000 g/mol are named PEO [99]. PEO is a hydrophilic synthetic polymer with low toxicity, low melting point, and the ability to interact with polarized surfaces [100,101]. In addition to water, PEO is soluble in different organic solvents such as chloroform, ethanol, and dimethylformamide [102]. Besides being biodegradable and biocompatible, its low cost makes this biomaterial even more attractive [103]. PEO is usually associated with other polymers in order to improve their mechanical properties and spinnability, the most common ones being chitosan (CHT) and gelatin (Gel) [104,105].

Cui et al. successfully developed continuous polyelectrolyte complex fibers, PEC, (chitosan/polystyrene sulfonate, chitosan/poly (acrylic acid), chitosan/poly (vinyl sulfate)), as well as PEO-doped composite fibers in a water-based wetspinning process, by interfacial complexation within a core–shell spinneret. They concluded that the addition of PEO to the spinning solutions proved to be an effective method to improve the elongation at break, break stress, and swelling behavior of PEC fibers. Tensile tests demonstrated an 80% increase in Young’s modulus as well as a 170% increase in tensile strength. The biocompatibility and suitability of CHT/PSS-PEO fibers for TE applications was also confirmed by a four-day cultivation of human HeLa cells on PEO-doped PEC scaffolds. A cell viability of 81.49 ± 2.95% was seen in the tissue structures by a subsequent analysis with fluorescence-based live/dead assay (Figure 7a,b). The suitability of the CHT/PSS-PEO fibers as a scaffold for cell culture applications was also demonstrated with the formation of focal adhesions and the orientation of the actin filaments [106].

Table 5 summarizes the most important works where PEG and PEO were used for biomedical applications.

## 6. Incorporation of Reinforcing Agents

As mentioned before, the right combination of biodegradable polymers gives rise to wetspun fibrous systems for tendon and ligament treatment. However, most of the systems have poor mechanical properties. To enhance these properties, without compromising biological performance, cellulose nanocrystals (CNC) have been extensively used as a reinforcement material [27,28]. The microfibrous structures can also be functionalized using nanoparticles to give them new physical and chemical properties, such as increasing the strength and endurance, or improving anti-microbial, anti-adhesive, and anti-inflammatory properties [29,30,31,32,33,34]. The addition of growth factors may also be necessary to obtain suitable structures for an adequate regeneration of the tissue [2].

### 6.1. Cellulose Nanocrystals (CNC)

Cellulose is the most abundant biopolymer in nature and can be found in algae, plants, and some bacteria [109]. Due to the exceptional versatility of these nanoscale materials, the use of nanocelluloses in the production of new fully biodegradable and bio-based materials has become a fast-growing area of interest in many fields [27,110]. In addition to their natural abundance, sustainability, recyclability, crystallinity, high surface area, good mechanical properties, and inherent biocompatibility, among others, nanocelluloses can be implemented not only in niche applications such as food, sensors, pharmaceuticals, household products, and cosmetics, but also in biomedical applications [27,111].

Nanocellulose is derived from cellulose; however, because nanocelluloses range from short, rod-shaped particles (CNCs) to long, flexible fibrils/ribbons (cellulose nanofibrils (CNFs) and bacterial cellulose (BC)), the choice of which type to use in which structure is fully dependent on the desired properties of the final material [27,110]. BC is produced by bacteria; synthesized inside them and expelled through their pores, but its large-scale production is still limited, due to the high cost of maintaining bacterial growth and low yield [110]. The main differences between CNCs and CNFs are their dimensions and crystallinity. CNFs are produced through mechanical or chemical treatments, and are mixtures of amorphous with crystalline chains, with lengths of up to several micrometers. On the other hand, CNCs are produced through chemical treatments, usually by acid hydrolysis of cellulosic materials disrupting the hydrogen bonds, in which the crystalline zones are extracted and preserved, while the amorphous ones are removed. These highly crystalline structures generally have lengths below 500 nm [112]. The structure and morphology of the final nanocrystals depends on the origin of the cellulose, the extraction method and conditions, the types of acids during the hydrolysis, and also the drying technique used [110,111,112,113,114]. Both CNCs and CNFs have immense potential for various applications, however this study focuses on the use of CNCs due to their excellent properties associated with their nanometer size.

Within the last decade, with their commercialization, the incorporation of reinforcement nanocellulose products have received outstanding attention from researchers. Besides their environmental benefits, such as biodegradability and renewability, cellulose nanoparticles also possess a high aspect ratio, a large surface area, a very low specific density, and high strength and modulus compared to conventional fibers [110]. Thus, as mentioned before, the incorporation of these reinforcing agents into polymeric systems makes it possible to improve the mechanical properties of fibers while ensuring the development of a biodegradable and fully bio-based system without compromising biological performance. Sheng et al. successfully produced CNC/PCL nanocomposite fiber mats via electrospinning. The produced mats were extensively characterized by scanning electron microscopy, differential scanning calorimetry (DSC) analysis, thermogravimetric analysis (TGA), uniaxial tensile tests, and static water contact-angle analysis. The results showed that the mechanical properties and thermal stability of the nanocomposite fiber mats were improved, the electrospun fiber diameter decreased, the tensile modulus and strength improved, and the surface hydrophilicity increased in comparison with neat PCL fiber mats. Thus, it was shown that the CNCs were relatively good reinforcing fillers for PCL [115]. Moreover, Xiang et al. incorporated CNCs prepared from microcrystalline cellulose into electrospun PLA fibers. It was reported that the strength of the electrospun non-woven fabrics was improved by 30% with 1 wt% loading of CNCs, which acted as a nucleating agent of PLA crystallization and hence increased the crystallinity of PLA in the resulting nanocomposite fibers [70].

In fact, several studies have demonstrated the successful incorporation of this reinforcing agent in different polymeric systems for the production of functionalized fibers, especially by wetspinning [71,116,117]. Liu et al. prepared sodium alginate/CNC fibers using a wetspinning method (Figure 8). The aim was to enhance the mechanical strength of sodium alginate fibers. Therefore, the structure and mechanical properties of sodium alginate/CNC produced fibers were characterized by several techniques. Figure 9 shows the stress–strain curves of the obtained fibers. As expected, the incorporation of CNC significantly improved the strength of the alginate fibers because of the uniform distribution of the crystals in the alginate matrix. The tensile strength and elongation at break of the alginate fibers increased with increasing CNC content from 0 to 2 wt%. The presence of a small amount of CNC largely improved the tensile properties of the fibers [118].

Uddin et al. conducted a study into CNC prepared from native cotton that was incorporated into PVA as reinforcing fillers. Homogeneous dispersions of PVA–CNC, with different CNC fractions, were successfully spun into composite fibers through wetspinning. The composite fiber with a small amount of CNC (5 wt%) showed higher drawability compared to the neat PVA fibers. These fibers also exhibited extremely highly orientated CNC and excellent mechanical properties. Therefore, the authors found that adding CNC not only increased the orientation of the crystal fraction of PVA but also improved the modulus and the strength of the fibers. Thus, with nanocellulose having no cytotoxicity, the nanocomposites prepared in this work are believed to be fully biodegradable and biocompatible even though no biological tests were made [109,116].

### 6.2. Nanoparticles

Tendons, ligaments, and associated ECM are composed of nanostructured materials. For this reason, in recent years there has been a growing interest in developing novel nanomaterials for tendon regeneration. Nanoparticles (NPs) are a material with dimension of <100 nm. They represent a bridge between structures at the atomic level and materials of conventional size. NPs can be exploited in different ways to improve tendon and ligament healing and regeneration, ranging from scaffold manufacturing (increasing the strength and endurance or increasing anti-adhesive, anti-microbial, and anti-inflammatory properties) to gene therapy [29,119]. These new properties would not be possible if using macrosized particles in the polymer matrix [119]. Furthermore, NPs are very attractive materials to be used in the development of functional fibrous systems, due to their higher surface area and exclusive electrical, optical, and catalytic properties [31,120].

Kwan et al. used Achilles Sprague-Dawley rats to study the effects of silver nanoparticles (AgNPs) in vitro and in vivo during the early stages of the tendon healing process. The in vitro analysis demonstrated that AgNPs encouraged primary tenocyte proliferation and collagen synthesis. The tensile modulus of the group treated with nanoparticles was much higher than that of the untreated group, but considerably lower than that of the normal tendon, according to biomechanical testing on the 42-day post-op Achilles tendon. The addition of silver nanoparticles accelerated the tendon healing process and altered the ECM composition, giving more and higher-quality collagen fibrils. According to the study’s findings, AgNPs promote cell proliferation and the synthesis of collagen and proteoglycans, which are all beneficial for Achilles tendon recovery. The silver nanoparticles also showed a reduction in the inflammatory response, minimizing fibrotic healing and scarring [121]. As said before, a study by Calejo et al. evaluated the development of gradient scaffolds to replicate the tendon-to-bone interface using aligned microfibers produced by wetspinning (Figure 6a). For this purpose, two formulations were considered: PCL/gelatin, to mimic tendon tissues, and PCL/gelatin incorporating nano-to-microsized hydroxyapatite (HAp) particles, to mimic bone (Figure 6b). Interestingly, the difference in composition between the two formulations was reflected in the properties of the obtained fibers. Biological performance demonstrated the synthesis of collagen type III by human adipose-derived stem cells (hASCs) on both scaffolds. However, in the presence of HAp, biological studies revealed that the fibers were not only able to support cell proliferation but also to favor cellular anisotropic alignment or to induce an osteogenic-like phenotype in human adipose-derived stem cells (hASCs). After an extensive characterization, it was concluded that through the addition of HAp into the initial polymeric solutions, structural and compositional properties could be tuned [58].

Recently, metal oxide NPs have emerged as a great alternative for the development of multifunctional fibrous systems [33,34]. Due to the excellent chemical and physical properties of these NPs, they are attracting more and more attention. B. Kolathupalayam Shanmugam et al. prepared titanium dioxide (TiO_2_) nanoparticle-reinforced chitosan-sodium alginate nanocomposites by solvent casting. The aim of the study was to synthesize metal oxide-based biomimetic nanocomposites to overcome the risk associated with artificial bone TE. In vitro biocompatibility of the prepared TiO_2_ was confirmed through hydroxyapatite (HAp) formation. TiO_2_ chitosan-sodium alginate composites also exhibited better protein absorption capability, suitable for cell attachment and growth, as well as better antibacterial activity against Escherichia coli and Staphylococcus aureus. In addition, better cytotoxicity against osteosarcoma (MG-63) cell lines and anti-proliferative activity against human bladder tumor (UC6) cell lines ensured suitability for bone tissue engineering. In general, the composite scaffolds containing TiO_2_ nanoparticles demonstrated better properties for bone tissue engineering applications [122]. Araújo et al. functionalized flax fabrics with MgO NPs and MgO–SiO_2_ core–shell NPs by a simple and sustainable in situ synthesis. The synthesis of the isolated NPs was also tested. MgO NPs and MgO–SiO_2_ core–shell NPs were successfully synthesized as well as the bonding between the MgO core and the SiO_2_ shell. The flax fabrics were also successfully functionalized with both NPs, and a homogenous distribution all over the fabric’s surface was observed. It was verified that a core–shell structure with SiO_2_ helped to better anchor the MgO NPs onto the fibers’ surface since the MgO–SiO_2_ core–shell NPs were more attached to the fabrics than the MgO ones. More importantly, the final samples presented multifunctionality with anti-bacterial activity, hydrophobicity, UV protection capacity, and methylene blue (MB) degradation capability. As seen in Figure 10, the samples presented a bactericidal effect against S. aureus and a bacteriostatic effect against E. coli. Overall, the work shows that these NPs are excellent candidates for a wide range of applications such as tendon TE [34].

### 6.3. Growth Factors

Recently, one of the most commonly proposed approaches to achieve tissue regeneration is the use of several growth factors simultaneously [123]. This approach is based on the fact that growth factors are not naturally synthesized in an independent way, but in the form of “cocktails” working together during the tendon repair process [124]. Although there is more knowledge about their synthesis, structure, and how they work, there is still a lack of knowledge on how to use them in TE, especially when more complex approaches are proposed. In addition, due to their limited half-life in vivo, the direct and local delivery of growth factors has limited use [123]. Therefore, there is a need for more advanced strategies for a sustained, safe, and reproducible delivery. Studies in which growth factors have been incorporated into scaffolds for tendon healing have shown promising results, allowing a more sustained and controlled release.

Various growth factors, such as PDGF-BB (Platelet Derived Growth Factor-BB) [125], IGF-1 (Insulin-like Growth Factor 1) [126], BMP-7 (Bone Morphogenetic Protein 7) [127], BMP-12 (Bone Morphogenetic Protein 12) [128], VEGF (Vascular Endothelial Growth Factor) [129], SDF-1α (Stromal cell-Derived Fac- tor 1) [130], bFGF (basic Fibroblast Growth Factor) [131], TGF-β1 (Transforming Growth Factor beta1) [132], and TGF-β3 (Transforming Growth Factor beta3) [133] have shown to be of great importance for tendon tissue engineering [2,123]. Table 6 summarizes some of the studies that involve a few of these growth factors as well as the type of injury and the main results obtained.

## 7. Spinning

It is known that fibrous scaffolds created with different fiber-based fabrication techniques are suitable for replacing anisotropic tissues and accelerating their healing. Wet- and electro-spun constructs are the most often used fiber-based scaffolding technologies for tendon and ligament tissue regeneration [1]. In recent years, electrospinning techniques have also been used to process biodegradable polymers, such as PHAs, intended for biomedical applications [134]. The relationship between processing parameters and electrospun fiber assembly architecture was examined in one of the earliest publications on electrospinning of PHA, which was published in 2006 [135]. The investigation led to the creation of a number of scaffolds with an average fiber diameter of a few microns, made of PHB, PHBV, or a combination of both (75:25, 50:50, or 25:75 weight ratio). A phase inversion mechanism associated with the swift evaporation of the solvent used (chloroform) was proposed by the authors as the explanation for the rough surface that SEM analysis of the PHB/PHBV blend fibers revealed. These scaffolds sustained the in vitro growth of mouse fibroblasts (L929) and human osteoblasts (SaOS-2), at higher levels than analogous cast-films, indicating biocompatibility of these materials in both types of cells. As reported, the viability of the cells cultured with an extraction medium is reported in Figure 11, in terms of the relative absorbance with respect to the absorbance value of SaOS-2 that were cultured with fresh SFM for the same culture period [136].

Chen et al. fabricated a bioactive cartilage tissue engineering scaffold that promotes cartilage regeneration. In this study, electrospun PHBV fibrous scaffolds were modified with QUE to enhance the bioactivity of PHBV. The modified PHBV fibrous scaffolds promoted the proliferation of chondrocytes, maintained the chondrocytic phenotype, and facilitated the formation of cartilage ECM. More importantly, the PHBV-g-QUE fibrous scaffold significantly promoted maturation of neo-cartilage tissue and cartilage regeneration in vivo compared with the neat PHBV fibrous scaffold (Figure 12). The results suggested that the PHBV-g-QUE fibrous scaffold can potentially be applied in cartilage tissue engineering (CTE) [137].

Wu et al. successfully developed tailored nanofibrous scaffolds of hydroxyapatite (HAp) dispersed in a polycaprolactone/chitosan (HAp-PCL/CHT) nanofibrous matrix for tendon and ligament TE by the electrospinning process. Favorable mechanical properties (load and modulus), cellular responses, and biocompatibilities were achieved. The load and modulus of the produced HAp-PCL/CHT fibers were 250.1 N and 215.5 MPa, which is very similar to the standard value of the human tendon and ligament tissues. The cellular responses and biocompatibility of the nanofibrous scaffolds were investigated with human osteoblast (HOS) cells and the microscopic images clearly showed that the HOS cells were well attached and flatted on the scaffolds (Figure 13a,b,d). It was also shown that the HAp dispersed PCL/CHT nanofibrous scaffolds promoted higher adhesion and proliferation of HOS cells, comparable to the nanofibrous scaffolds without HAp nanoparticles (Figure 13c). Overall, the physical and biological properties of the synthesized HAp-PCL/CHT scaffolds were very close to that of the normal human tendon and ligament [87].

Moreover, Domingues et al. enhanced the biomechanical performance of anisotropically aligned electrospun nanofibrous scaffolds based on poly-3-caprolactone/chitosan (PCL/CHT) by incorporating small amounts of CNC (up to 3 wt%). The aligned PCL/CHT/CNC nanocomposite fibrous scaffolds met not only the mechanical requirements for tendon TE applications but also provided tendon mimetic extracellular matrix (ECM) topographic cues, which is a key feature for maintaining the tendon cell’s morphology and behavior [28].

In a study made by Toncheva et al., two types of antibacterial micro- and nano-fibrous mats based on PLA and PEG were prepared by electrospinning. PEG incorporation was achieved by its physical blending with, or chemical grafting on, PLA. Physical blending or chemical grafting showed a plasticizing effect on PLA but did not significantly modify the wettability of the mats. The PEG incorporation method also influenced the mechanical properties of the mats (Table 7). Fibrous mats of physically blended PLA and PEG proved quasi-ductile behavior, while brittle behavior was registered for the chemically grafted mats [138].

In another study by Huang et al., PCL/PEO nanofibers with or without Dipsacus asper Wall extracts (DAE) were produced under optimal electrospinning conditions with the aim of evaluating the osteogenic differentiation of periodontal ligament stem cells (PDLSCs) of DAE. The physical property analysis of the obtained nanofibers included Fourier transform infrared spectroscopy (FTIR), mechanical strength, biodegradability, swelling ratio and porosity, and cell compatibility. The results of the study confirmed that both DAE and PCL/PEO nanofibers have the effect of promoting osteogenic differentiation. However, the addition of DAE effectively increased the elongation, swelling, porosity, and degradability of the PCL/PEO nanofibers, resulting in fibers with better applicability as tissue engineering scaffolds and increased osteogenesis induction effects [139].

Overall, the electrospun nanofibrous constructs are also possible options to create scaffolds for tendon and ligament regeneration. They offers a suitable environment for biosignaling, support proper cell attachment and proliferation, and prevent the adhesion of surrounding tissues. However, poor cell infiltration and a shortage of cell binding sites are still thought to be the technologies’ principal drawbacks [1].

Recently, multilayered systems have appeared as the most promising candidate for tendon and ligament tissue engineering. By combining the characteristics of various source materials, morphologies, and production techniques, this method creates composite structures that may be the most advantageous in terms of biological response as well as mechanical and structural capabilities [1]. Lately, systems for tendon and ligament engineering that combine electrospun fibrous mats with woven/knitted structures in multilayered scaffolds have been investigated. Rashid et al. created a patch intended for tendon repair with two exterior layers made of electrospun polydioxanone (PDO) fibrous meshes and PDO woven structure, and an inner layer of aligned electrospun PCL fibers. A sheep model used for in vivo testing revealed blood vessel development and infiltration of cells, primarily tendon fibroblasts, onto the electrospun layer. There were no reports of an exaggerated inflammatory response or surrounding tissue adhesion [140].

Additionally, 3D printing technology has been combined with the electrospinning technique to produce fibrous meshes layered with printed structures. Touré et al. directly electrospun poly (caprolactone)/poly (glycerol sebacate) (PCL/PGS) nanofibers for tendon/ligament applications onto 3D printed PCL/PGS constructs coated with bioactive glasses. In vitro biocompatibility was found to be increased in the presence of the bioactive glasses, and sufficient mechanical qualities, such as Young’s modulus, were also noted [141].

Layer-by-layer scaffolds, in which hydrogels are stacked between electrospun, knitted, or braided structures are also thought to be promising alternatives for the same purpose. Zhao et al. created a multilayered system with a sheet of fibrin gel supplied with a basic fibroblast growth factor (bFGF) and mesenchymal stem cells (MSCs) sandwiched between two poly (lactide-co-glycolide) (PLGA) knitted constructions for tendon repair. The scaffold that was implanted in a rat model, specifically in a critical-size Achilles tendon defect, showed adequate biomechanical characteristics and allowed the MSCs that were added to it to express tenogenic markers, which supported tissue repair and regeneration [142]. A 3D multi-layered composite scaffold, made of layers of hydrogel loaded with MSCs and coated with synthetic electrospun nanofibers, was proposed by Rinoldi et al. It was established that the PCL and polyamide 6 (PA6) electrospun matrix provided the mechanical qualities and supported the entire build. The hydrogel layers, however, imitated a microenvironment favorable for cell encapsulation and proliferation since they were made of gelatin methacryloyl (GelMA) and alginate. In a specially designed bioreactor, mounted constructs were cultivated while receiving mechanical and biochemical stimulation. To encourage the tenogenic differentiation of MSCs, the concentration of the BMP-12 addition was optimized. Results obtained in vitro demonstrated the stimuli’s beneficial effects on tenogenic differentiation, alignment, and proliferation [128]. The findings indicated that the suggested structure may be employed to design functional tendons.

On the other hand, wetspun fibers have a direct correlation between yarn dimension and ultimate stress values, however they sometimes lack the mechanical properties to match tendon/ligament performance. These fibers have therefore frequently been braided or combined with other technologies in order to create structures with mechanical properties similar to those of native tissues [1]. Aiming to treat tendon and ligament injuries, Majima et al. developed a scaffold consisting of chitosan and hyaluronic acid. The wetspinning technique was used to create the polymer fibers, and the 3D scaffold was made using a braiding machine. While in vivo implantation in a rat model produced little toxicity and inflammation responses, mechanically maintaining the joint during the healing process, in vitro tests demonstrated collagen I and III deposition on the chitosan-hyaluronan wetspun scaffolds [143]. In a study by Guner et al., a dual-phase fibrous scaffold was created with the goal of enhancing the construct’s potential for healing by merging fibrous mats made by rotary jet spinning (RJS) and wet electrospinning (WES). Aligned PCL fibers (Shell) made by RJS and randomly oriented PCL or PCL/gelatin fibers (Core) made by WES systems were combined to create dual-phase scaffolds. The scaffolds replicated the remodeling and repair phases of tendon healing. Studies conducted in vitro revealed that the aligned PCL fiber shell of the dual-phase scaffold with randomly oriented core fibers boosted the initial adhesion and viability of cells. The created FSPCL/ESPCL-Gel 3:1 scaffold (FS = centrifugal force spinning/RJS, ES = wet electrospinning, Gel = gelatin) supported tenogenic differentiation, maintained high mechanical strength, and enhanced cell viability and orientation. The combination of two distinct fiber production methods produced scaffolds that were reliable and repeatable and could be used for tendon tissue engineering applications [144].

The main difficulty in tendon and ligament tissue engineering is the creation of a single structure with gradient mineralized and nonmineralized sections, different collagen interlacing, and hard-to-soft mechanical qualities [1]. To be more precise, tendon/ligament, fibrocartilage interface, and bone are the three distinct tissue regions that should be constructed. In the aforementioned study by Calejo et al., gradient scaffolds were developed to replicate the tendon-to-bone interface using aligned microfibers produced by wetspinning. Two formulations were taken into account for this use: PCL/gelatin to resemble tendon tissues and PCL/gelatin with nano-to-microsized hydroxyapatite (HAp) particles to simulate bone. The braiding of PCL/gelatin with PCL/gelatin/HAp microfibers resulted in a 3D gradient structure with a continuous topographical and chemical gradient, Figure 14. Overall, the findings strongly supported the viability of adopting straightforward fiber processing methods, such as wetspinning, to precisely regulate the topography and composition of fibers while tailoring cellular responses. The construction of 3D fibrous scaffolds for tendon-to-bone regeneration was also made possible by the integration of modern textile processes [58].

All in all, combining approaches to create scaffolds with higher structural and biomechanical qualities is fairly common. With this approach, the scaffold reflects a higher degree of complexity that will more closely resemble a natural tendon or ligament [2].

## 8. Conclusions and Perspectives

Tendons and ligaments are complex tissues with very unique qualities and traits that are directly tied to their role in the body. Millions of people experience tendon and ligament damage every day as a result of the extreme strain these tissues must endure. Patients dealing with an injury that has a serious impact on their quality of life, as well as healthcare systems all over the world, are affected by this serious issue. One of the biggest problems with this kind of injury is how difficult it is to heal from it, which is primarily due to the aforementioned tendon and ligament complex characteristics. The natural regeneration of the tissues is a challenging process that frequently yields fibrous tissue that is non-functional, only partially functional, or weaker than it was before. Many other treatments have been suggested over the years to restore the damaged tissues. Currently, surgical and conservative treatments are both commonly utilized to treat this kind of damage. Treatments using drugs, the infiltration of cells, or growth factors are also used. Among them, the most widely used today is the combination of a surgical operation with early mobility exercises, in the case of a severe rupture. Unfortunately, these therapies are not always successful because tendon re-injuries happen frequently and sometimes it is not possible to restore the structure or functionality that the tendon had before the injury.

In order to improve tendon and ligament regeneration and to recover its functionality, the application of TE to this type of tissue has been proposed. TE approaches aim to overcome the disadvantages of the current treatments by stimulating the regeneration and formation of neotissues in order to obtain structures that resemble the original ones. Multiple materials can be used to obtain the desirable structures or scaffolds. Biodegradable polymers have drawn attention for this application, due to their biocompatibility and biodegradability. However, their use still presents some drawbacks. Therefore, to overcome these disadvantages, researchers started to consider synthetic polymers as reinforcements for naturally derived ones, exploring and designing different composites and multi-layered structures. In this way, it is possible to combine the biocompatibility and biological properties of natural polymers with the high mechanical strength and durability characteristics of synthetic polymers.

The technique used to obtain the scaffolds is also decisive in determining the properties of the final structure. Moreover, the selected materials are also linked to the type of technology used. Wetspinning is a controllable and simple manufacturing method that allow obtention of micron-sized fibers. Its application in tendon and ligament tissue engineering has been recently explored. One important advantage of this method is that it allows adjustment or control over the fibers diameter, the pore size, and the porosity of the final structure. In the presence of mild conditions, the control of the fiber characteristics make this technique suitable for cell incorporation, as cell adhesion and proliferation are facilitated. However, the main advantage of this technique for tendon and ligament tissue engineering is the obtention of fibrillar structures. A wide range of materials can also be processed using the wetspinning method, including natural derived polymers (chitosan, collagen, gelatin, etc.) as well as native ECM components.

Another element used in TE are reinforcing agents. The right combination of biodegradable polymers and the wetspinning technique gives rise to wetspun fibrous systems for tendon and ligament treatment. However, the neat polymeric fibrous structures present poor mechanical properties. To enhance these properties, without compromising biological performance, and also to give them new physical and chemical properties (increased strength and endurance or new anti-microbial, anti-adhesive, and anti-inflammatory properties), cellulose nanocrystals, nanoparticles and/or growth factors have been extensively used as a reinforcement material.

It can be extrapolated that strategies that combine the aforementioned factors show more promising outcomes. Additionally, the information being created in the sector indicates that it will be much simpler in the future to tailor the structures and therapies employed to each type of patient and injury. While tendon and ligament tissue engineering has made significant progress, the solutions suggested and the outcomes so far have only allowed for the creation of scaffolds with characteristics that are still too distinct from those of natural tissues. So far, hydrogels or scaffolds having a structural role have exhibited the best outcomes in terms of the structure.

Everything that has been said so far implies that TE used on tendons and ligaments will continue to progress in the years to come, toward more complicated techniques and structures that eventually manage to successfully and fully regenerate the injured tissues. This is especially true given the interest it is stirring in research organizations around the world. However, while we recognize the potential of TE in tendon and ligament regeneration, we should also consider the challenges that TE may face in this. These challenges include unknown toxicity, selection of the right combination (solution, NPs, growth factors, etc.), and translating results from preclinical studies to a clinical setting. Solving these problems will help TE improve the efficacy of fibrous scaffolds treatments with the aim of tendon tissue regeneration.

## Figures and Tables

**Figure 1 pharmaceutics-14-02526-f001:**
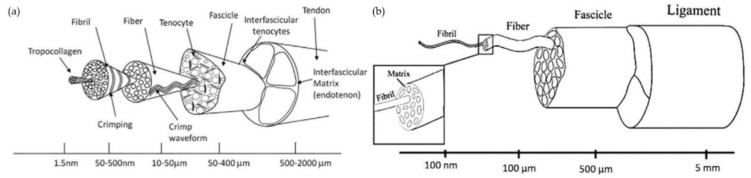
Physiology and mechanical characteristics of tendon and ligament tissue. (**a**) Tendon hierarchical structures. (**b**) Ligament hierarchical structures. Reprinted/adapted with permission from Ref. [1]. 2022, John Wiley and Sons”.

**Figure 2 pharmaceutics-14-02526-f002:**
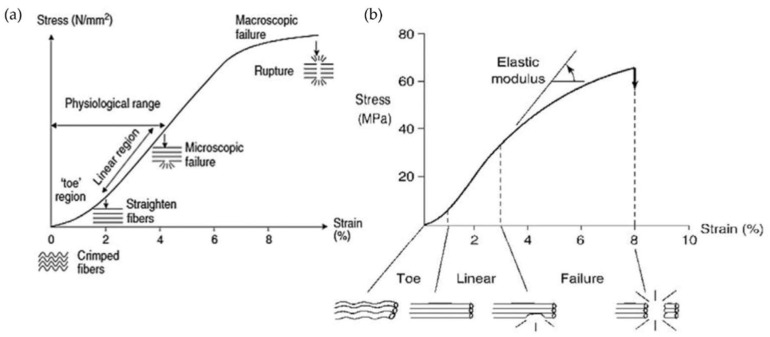
Mechanical characteristics of tendon and ligament tissue. (**a**) Representative stress–strain curve of tendon subjected to mechanical tensile. (**b**) Representative stress–strain curve of ligament tissue failed in tension. Reprinted/adapted with permission from Ref. [1]. 2022, John Wiley and Sons”.

**Figure 3 pharmaceutics-14-02526-f003:**
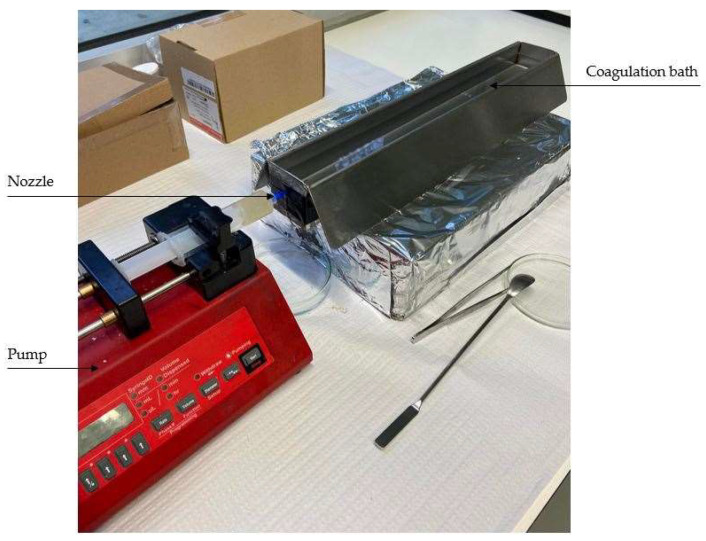
Example of a Wetspinning set-up.

**Figure 4 pharmaceutics-14-02526-f004:**
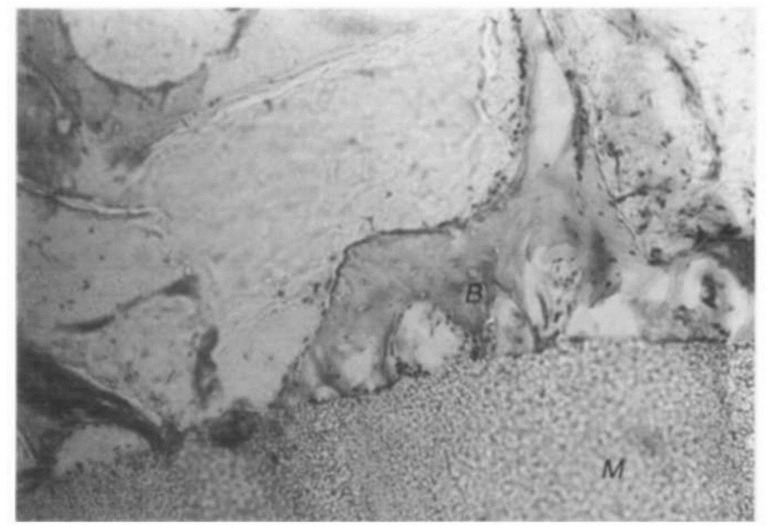
Interface histology after 3 months implantation showing bone growth (B) direct to the material (M). Reprinted/adapted with permission from Ref. [73]. 2022, Elsevier.

**Figure 5 pharmaceutics-14-02526-f005:**
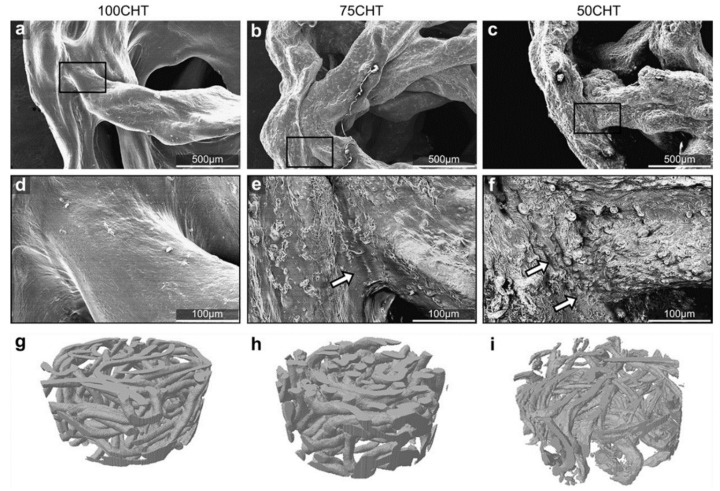
SEM microphotographs of the (**a**,**d**) 100CHT, (**b**,**e**) 75 CHT and (**c**,**f**) 50 CHT fiber-meshes after the thermal treatment at T_a_ = 60 °C and t_a_ = 3 h. The (**d**–**f**) images correspond to the magnification of the area delimited by the rectangular box on the (**a**–**c**) images; representative 3D μCT images of the (**g**) 100 CHT, (**h**) 75 CHT and (**i**) 50 CHT fiber-meshes. Reprinted/adapted with permission from Ref. [73]. 2022, Elsevier.

**Figure 6 pharmaceutics-14-02526-f006:**
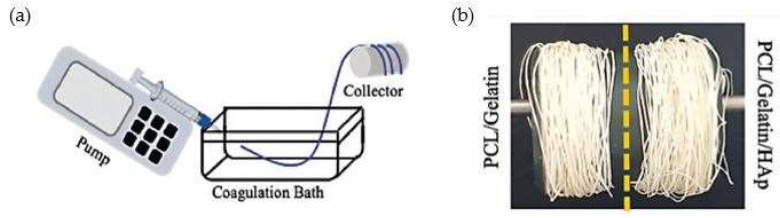
(**a**) Schematic representation of a continuous wetspinning set-up. (**b**) PCL/gelatin and PCL/gelatin/HAp wetspun fibers. Reprinted/adapted with permission from Ref. [58]. 2022, John Wiley and Sons.

**Figure 7 pharmaceutics-14-02526-f007:**
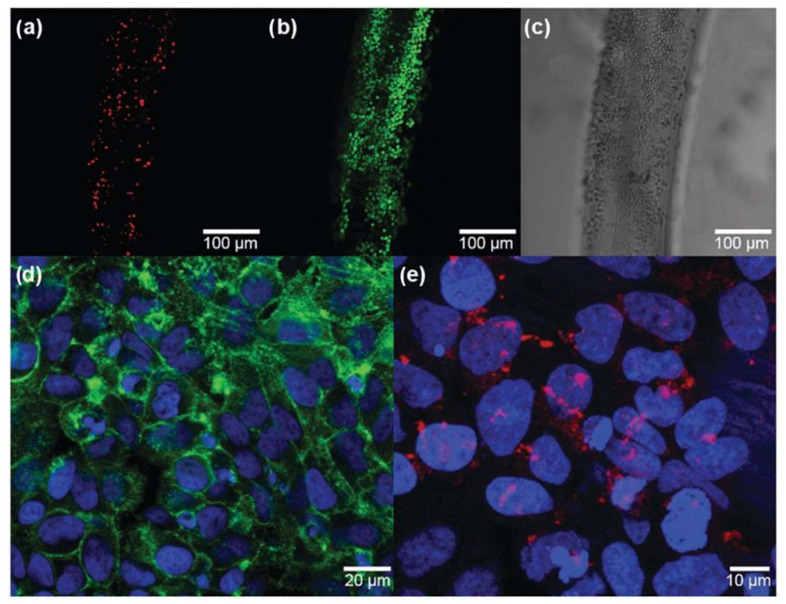
Human HeLa cells on a CHT/PSS-PEO fiber. (**a**) Dead cells labeled with ethidium homodimer-1 (EthD-1). (**b**) Living cells labeled with calcein. (**c**) Bright field image of a cell monolayer. (**d**) Visualization of the nuclei and actin cytoskeleton filaments by DAPI/Vinculin staining. Culture time: 4 d; medium: DMEM + 10% FBS + 1% PS; 37 °C humidified air 5% CO_2_. (**e**) Visualization of the nuclei and focal adhesions by DAPI/Vinculin staining. Culture time: 4 d; medium: DMEM + 10% FBS + 1% PS; 37 °C humidified air 5% CO_2_ [106].

**Figure 8 pharmaceutics-14-02526-f008:**
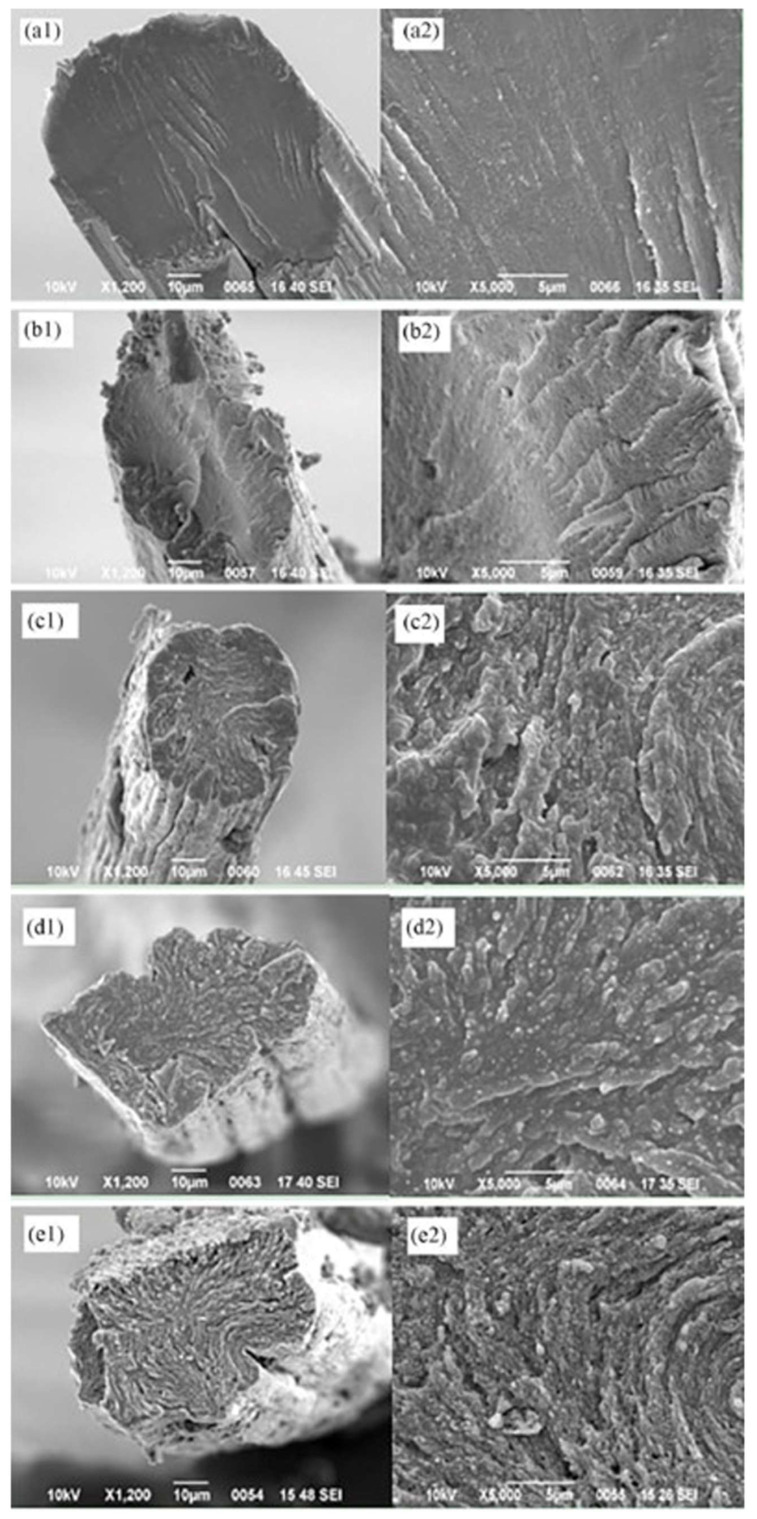
SEM images of the alginate fibers’ cross-sections with different CN contents: (**a1**,**a2**) 0 wt%, (**b1**,**b2**) 0.5 wt%, (**c1**,**c2**) 2 wt%, (**d1**,**d2**) 8 wt%, and (**e1**,**e2**) 16 wt% Adapted form [118], J. Eng. Fiber. Fabr. 2019.

**Figure 9 pharmaceutics-14-02526-f009:**
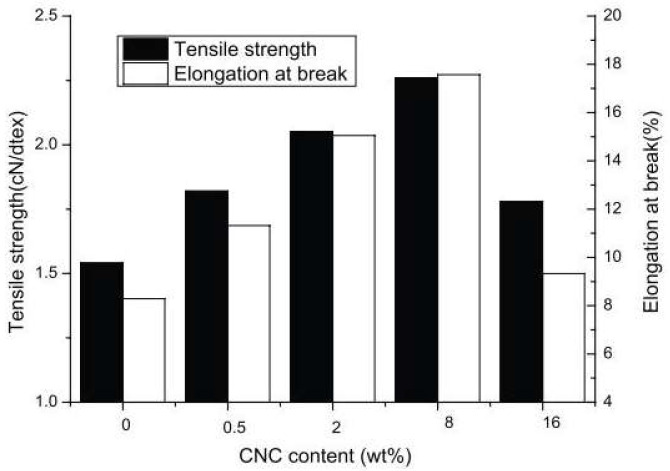
Mechanical properties of SA/CN fibers. Adapted form [118], J. Eng. Fiber. Fabr. 2019.

**Figure 10 pharmaceutics-14-02526-f010:**
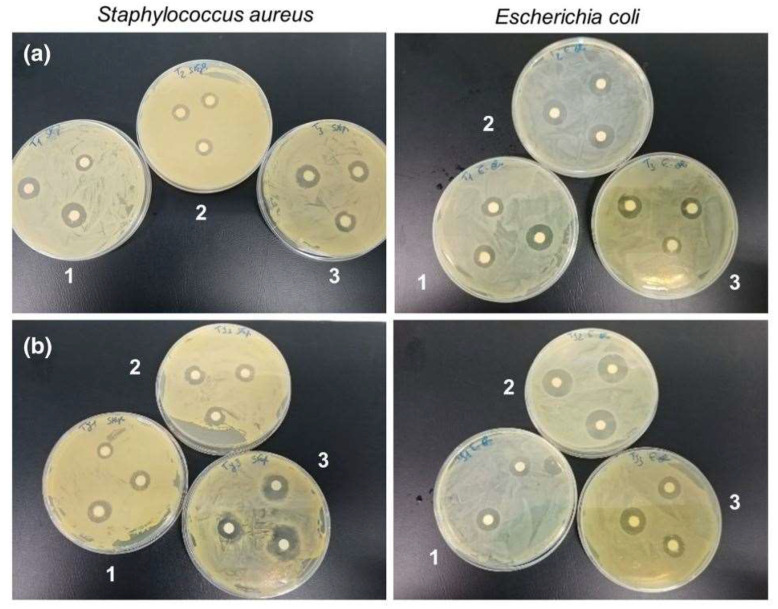
Comparison of the inhibition zone of (**a**) MgO NPs and (**b**) MgO–SiO_2_ core–shell NPs solutions against S. aureus and E. coli bacteria with different concentrations: (1) 0.0005 mg/mL, (2) 0.0010 mg/mL and (3) 0.0050 mg/mL. Reprinted/adapted with permission from Ref. [34]. 2022, Springer Nature.

**Figure 11 pharmaceutics-14-02526-f011:**
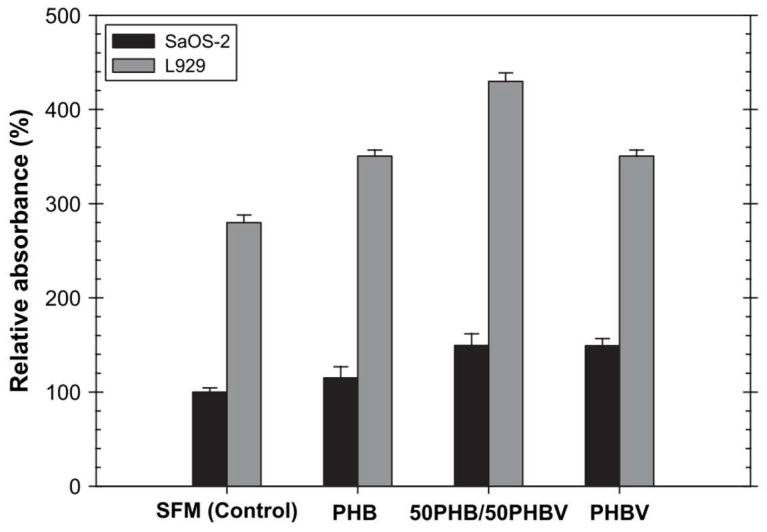
Indirect cytotoxicity evaluation of the as-spun PHB, PHBV, and PHB/ PHBV fiber mats based on the viability of human osteoblasts (SaOS-2) and mouse fibroblasts (L929). Reprinted/adapted with permission from Ref. [136]. 2022, Elsevier.

**Figure 12 pharmaceutics-14-02526-f012:**
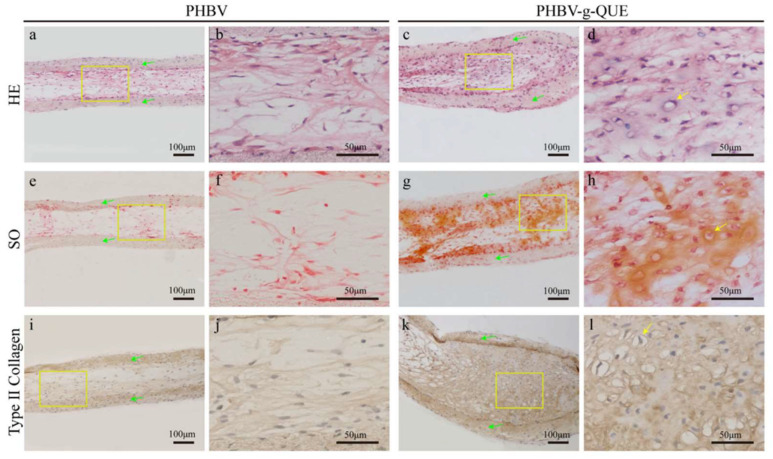
Histological analysis after the tested samples were implanted subcutaneously into nude mice for 4 weeks. (**a**–**d**) H&E staining, (**e**–**h**) SO staining, and (**i**–**l**) immunohistochemical staining of type II collagen. The green arrows indicate the scaffolds and the yellow arrows indicate the cartilage lacuna [137].

**Figure 13 pharmaceutics-14-02526-f013:**
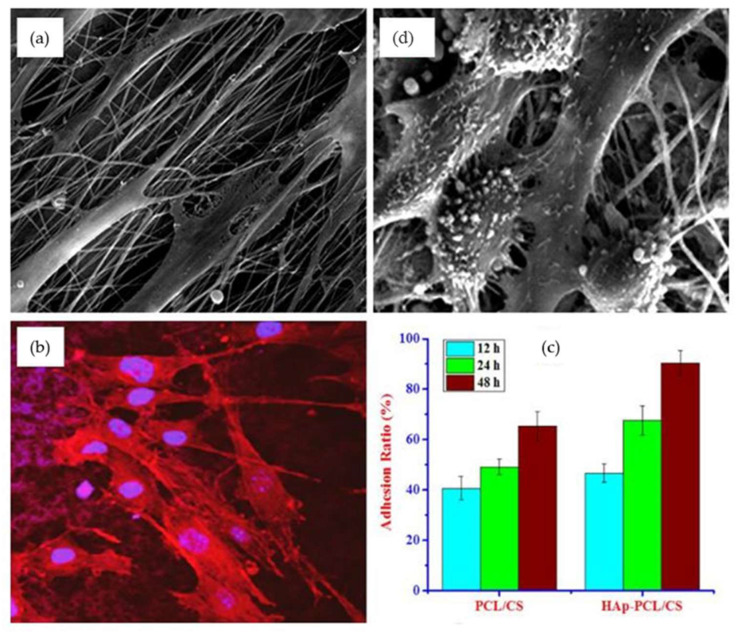
SEM and fluorescence images of HOS osteoblast attachment on PCL/CS (**a**) and HAp-PCL/CS (**b**,**d**) nanofibrous composite, (**c**) cell adhesion of the samples with incubation hours. Reprinted/adapted with permission from Ref. [87]. 2022, Elsevier.

**Figure 14 pharmaceutics-14-02526-f014:**
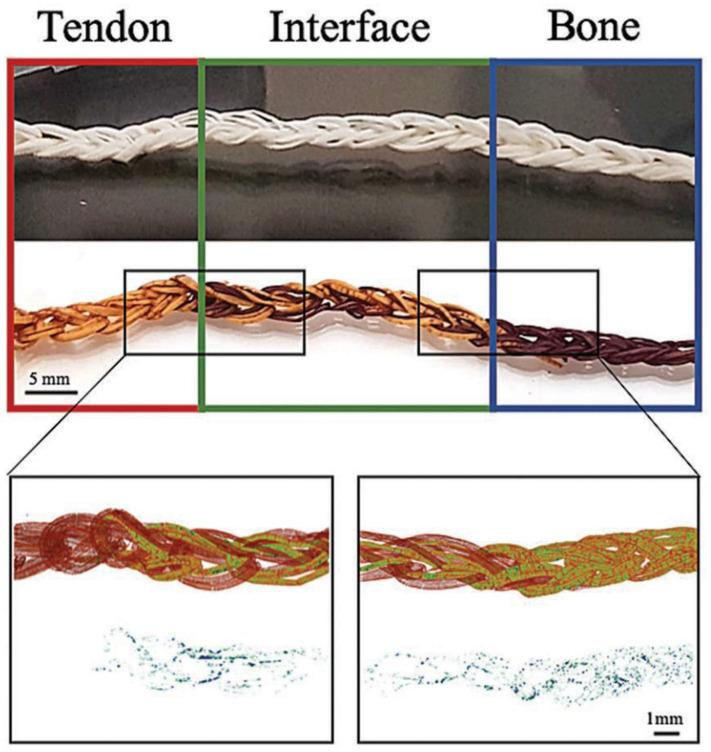
Morphology of produced scaffolds with HAp gradient. 3D scaffolds were produced by crochet using PCL/gelatin and PCL/gelatin/HAp microfibers to mimic tendon, interface, and bone. Scale bar, 5 mm. Reprinted/adapted with permission from Ref. [58]. 2022, John Wiley and Sons.

**Table 1 pharmaceutics-14-02526-t001:** Summary of the main advantages and disadvantages of some treatments used for tendon regeneration (Adapted from [2]).

Methods for tendon repair	Advantages	Disadvantages	References
Conservative treatment			
Extracorporeal shock wave therapy	Safe Simple to apply	Mechanism not fully understood The effects on healing are unclear	[40,41]
Ultrasound	Safe Simple to apply Short-term pain relief Adhesion prevention Reduction in the amount of inflammatory infiltrate	Mechanism not fully understood The effects on healing are unclear Too high-intensity ultrasound can have tissue destructive effects Cannot be used as a sole treatment, always used as a complement	[42,43]
Exercise	Decreased tendon volume Increased synthesis of type I collagen Improved tendon gliding and repair strength Adhesion prevention	Risk of re-injuring Most appropriate exercises for each type of injury are unknown	[44,45]
Surgical intervention	Alternative when other techniques have not worked or in cases of total tendon rupture Remove the damaged part of the tendon	Loss of tissue mechanical properties Generation of scar tissue or adhesions Unable to regenerate completely the injured tissue Risk of damage and infection	[46,47]
NSAIDs	Short-term analgesic effect	The effects on healing are unclear Negative effects when given postoperatively: decrease failure loads and increased rates of failure Gastrointestinal, cardiovascular and renal risks	[48,49,50]
Growth factors	Simplicity of the injection Well-studied targets Demonstrated effect on tendon regeneration Increasing load to failure and elongation values, stimulating collagen and matrix production, inducing tenocyte differentiation and tendon specific gene markers, inducing cell proliferation, etc.	Short half-life in the damaged tendon (requires repeated injections) Costly Little knowledge about the dosage and timing of injection	[44,51]
Cell therapy	Demonstrated effect on tendon regeneration Improved clinical outcome scores, improved biomechanical testing, increased collagen production and alignment, etc.	Difficult to maintain the cells at the specific site of injury Risk of unleashing an immune response	[52,53]
Gene therapy	Sustained and targeted production of growth factors and additional molecules Can avoid immunogenicity	Expensive Complicated manufacturing processes Poorly developed technique	[44,54]

**Table 2 pharmaceutics-14-02526-t002:** Set of parameters (solution, process, and environmental) that influence the diameter and final morphology of the fibers produced by the wetspinning technique.

Solution Parameters	Process Parameters	Environmental Parameters
Polymer(s) molecular weight	Coagulation bath (non-solvent)	Drying temperature
Polymer(s) concentration	Coagulation bath temperature	Humidity
Viscosity	Needle diameter	
Solvent	Flow rate	
	Washing bath	
	Dewatering bath	

**Table 3 pharmaceutics-14-02526-t003:** A summary of wetspun systems based on PHA for TE applications.

Polymers	Solvent/Mix of Solvents	Wetspinning Parameters	Application Perspective	Ref.
Polyhydroxybutyrate—98%/polyhydroxyvalerate 2% (PHBV)	Chloroform 20% (*w*/*v*)	Flow rate: 0.2 mL/min Coagulation bath: ethanol	Tissue Engineering	[75]
PHBV-O2-ELR-REDV (PHBV (HV content 8% *w*/*v*))	Chloroform 8% (*w*/*v*)	Flow rate: 1 mL/h Coagulation bath: methanol	Tissue Engineering	[76]
PHBHHx/PCL (12% *w*/*v*) PHBHHx/PCL weight ratios (3:1, 2:1 and 1:1)	Tetrahydrofuran (THF)	Coagulation bath: ethanol	Tissue Engineering	[83]
PHBHHx (25% *w*/*v*)	Chloroform	Flow rate: 0.2 mL/h Coagulation bath: ethanol	Tissue Engineering	[84]

**Table 4 pharmaceutics-14-02526-t004:** A summary of wetspun systems based on PCL for TE applications.

Polymers	Solvent/Mix of solvents	Wetspinning Parameters	Application Perspective	Ref.
PCL (10% (*w*/*v*)	Chloroform/tetrahydrofuran (3:1, *v*/*v*)	Flow rate: 1, 4, and 8 mL/h Coagulation bath: edible oil/hexane solution (3:1, *v*/*v*)	Tissue Engineering	[88]
PCL (6–20% *w*/*v*)	Acetone	Coagulation bath: methanol	Tissue Engineering	[89]
PCL (10–20% *w*/*v*) loaded with HNPs, CD or CD–HNP	Acetone Chloroform	Flow rate: 1.5–3.5 mL/h Coagulation bath: methanol, ethanol, water	Tissue Engineering and Regenerative Medicine	[90]
PCL (15% *w*/*v*)	Acetone	Flow rate: 2.25 mL/h Coagulation bath: ethanol	Tissue Engineering	[22]
CHT/PCL (100:0, 75:25, 50:50)	Formic acid (100 vol.%)	Coagulation bath: methanol	Tissue Engineering	[91]
PCL/gelatin (70:30) PCL/gelatin/HAp	PCL: formic acid/acetic acid (3:1 *v*/*v*) Gelatin: 80% acetic	Coagulation bath: glutaraldehyde/water (1.25% (*v*/*v*)) Flow rates: 0.25, 0.5 and 1 mL/h	Tissue Engineering	[58]

**Table 5 pharmaceutics-14-02526-t005:** A summary of wetspun systems based on PEG and PEO for TE applications.

Polymers	Solvent/Mix of Solvents	Wetspinning Parameters	Application Perspective	Ref.
Collagen I/PEG 4000 (3.25 wt%)	Acetic Acid	Flow rate: 0.01–5 mL/min Coagulation bath: 10% (*w*/*v*) PEG 20,000, 6.86 mg/mL TES, 4.14 mg/mL NaH_2_PO_4_, 7.89 mg/mL NaCl and 12.10 mg/mL Na_2_HPO_4_	Tissue Engineering	[98]
CNC-g-PEGX (X refers to the molecular weight of PEG)/sodium alginate (4 wt%)	Water	Coagulation bath: calcium chloride	Medical Textiles	[107]
Collagen (0.8–1.6% wt/vol) Crosslinked with PEG	Acetic Acid	Flow rate: 12 mL/h Coagulation bath: ethanol	Tissue Engineering	[108]
CHI/PSS-PEO (5 wt%)	---	Washing bath: 50:50 mixture of water and ethanol	Tissue Engineering	[106]

**Table 6 pharmaceutics-14-02526-t006:** Summary of some of the main growth factors used in tendon TE (Adapted from [2]).

Growth Factor	Injury	Main Results	Ref.
PDGF-BB	Achilles tendon full laceration	Increased tensile stress, failure stress, stiffness, and elastic modulus of treated tendons Upregulated expression of collagen I and III Thickening and enlargement of the cross-sectional area of the tendons Decreased cellularity	[125]
IGF-1	General tendon injury	Increased cellularity Increased collagen synthesis in TDSC Glycosaminoglycan synthesis	[126]
BMP-7	Rotator Cuff tear	Favorable orientation of rotator cuff collagen fibers Helps tendon-to-bone maturing Improves the ultimate force-to-failure	[127]
BMP-12	General tendon injury	Increased cell proliferation Upregulated expression of key tenogenic markers (collagen I, tenomodulin, scleraxis, TN-C)	[128]
bFGF	Achilles tendon defect	Increased tendon- related gene expression Improvement of the biomechanical strength Stimulate MSCs tenogenic differentiation	[131]

**Table 7 pharmaceutics-14-02526-t007:** Mechanical properties of the electrospun materials Reprinted/adapted with permission from Ref. [138]. 2022, European Polymer Journal.

Mat	Young’s Modulus (MPa)	Breaking Load (MPa)	Maximum Deformation (%)
PLA	94 ± 8	3.9 ± 0.03	56 ± 7
PLA/PEG_400_	150 ± 19	1.5 ± 0.02	35 ± 4
PLA/PEG_2000_	131 ± 16	0.8 ± 0.03	40 ± 5
PLA-*g*-poly(acrylIPEG)	155 ± 18	1.4 ± 0.02	4.6 ± 0.4
PLA/5N8Q	65 ± 5	3.0 ± 0.06	50 ± 8
PLA/ PEG_400_/5N8Q	83 ± 6	0.8 ± 0.08	21 ± 2
PLA/ PEG_2000_/5N8Q	88 ± 7	0.6 ± 0.04	25 ± 3
PLA-*g*-poly(acrylIPEG)/5N8Q	105 ± 13	0.8 ± 0.03	3.7 ± 0.3

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
