# Peer review of "Wetspun Polymeric Fibrous Systems as Potential Scaffolds for Tendon and Ligament Repair, Healing and Regeneration"

_pharmaceutics, 2022, doi:10.3390/pharmaceutics14112526_

Round 1

Reviewer 1 Report

General comments:

In this review, the authors just summarize the topic with a particular bias towards the wet-spun polymeric fibrous systems. The topic is of importance.

The wet-spinning technique provides a possibility of fabricating cell-laden highly biocompatible fibrous constructs is most probably the greatest advantage of this technique which avoids the use of toxic solvent during the spinning process.

However, the mechanical properties of the wet-spun polymeric systems are still considered insufficient for tendon applications. The authors should provide some comments on the point with advantages and limitations.

Regarding multilayered scaffolds

Recently, the electrospinning technique has also been combined with 3D printing technology, creating fibrous meshes layered with printed structures.

Additionally, layer-by-layer scaffolds are also considered promising alternatives for reproducing the native characteristic of tendon/ligament tissues.[

The authors should provide some comments on the point.

Regarding Engineering of tissue interfaces

The main difficulty in this field is the design of gradient scaffolds with zonal structure, architectures, compositions, and mechanical properties. However, this aspect is crucial to develop systems which can guarantee the physiological bio-functionality of the tissue.

The authors should provide some comments on the three different tissue regions: tendon/ligament, fibrocartilage interface and bone as well as the design of the tendon–muscle interface.

My concern is the lack of combination of different techniques in this review.

The combination of different techniques is considered crucial to produce scaffolds which can recapitulate the native tissue properties and promote tissue regeneration. The multilayered systems, where each compartment can be independently produced and tailored, have emerged as the most effective alternatives for tendon and ligament TE.

I recommend the author to take into account the above considerations, which might help for improving the overall quality of the manuscript.

Author Response

Dear Editor and Dear Reviewers,

Please find enclosed the revised version of the above referenced manuscript. The authors wish to thank all the reviewers for their valuable comments and inputs, having reviewed and edited the original manuscript accordingly, while highlighting all major changes (in yellow).

This document also contains a detailed, point to point answer to all the reviewers’ remarks and indications, after this cover letter.

The authors also had into consideration the Editor comments about some paragraphs containing similarities with other works, in that sense several paragraphs were rewritten and corrected as requested.

Looking forward to hearing from you,

 Yours sincerely,

Doctor Diana P. Ferreira

Researcher at Minho University

All reviewers’ comments have been acknowledged and analysed. This document contains a point-to-point answer to each comment, while mentioning any relevant edits to the manuscript.

Reviewer #1:

Comment #1: The wet-spinning technique provides a possibility of fabricating cell-laden highly biocompatible fibrous constructs is most probably the greatest advantage of this technique which avoids the use of toxic solvent during the spinning process. However, the mechanical properties of the wet-spun polymeric systems are still considered insufficient for tendon applications. The authors should provide some comments on the point with advantages and limitations.

Author’s reply:

The authors agree with the reviewer and add the suggested work as more examples in this area, as well as more citations. 

The following text was added to the manuscript:

Line 872:

On the other hand, wetspun fibres have a direct correlation between yarn dimension and ultimate stress values, however they sometimes lack the mechanical properties to match tendon/ligament performance. These fibres have therefore frequently been braided or combined with other technologies in order to create structures with mechanical properties similar to those of native tissues [1]. Aiming to treat tendon and ligament injuries, Majima et al. developed a scaffold consisting of chitosan and hyaluronic acid. The wetspinning technique was used to create the polymer fibres and the 3D scaffold was made using a braiding machine. While in vivo implantation in a rat model produced little toxicity and inflammation response, mechanically maintaining the joint during the healing process, in vitro tests demonstrated collagen I and III deposition on the chitosan-hyaluronan wetspun scaffolds [143]. In a study by Guner et al., a dual-phase fibrous scaffold was created with the goal of enhancing the construct's potential for healing by merging fibrous mats made by rotary jet spinning (RJS) and wet electrospinning (WES). Aligned PCL fibers (Shell) made by RJS and randomly oriented PCL or PCL/gelatin fibers (Core) made by WES systems were combined to create dual-phase scaffolds. The scaffolds replicated the remodeling and repair phases of tendon healing. Studies conducted in vitro revealed that the aligned PCL fiber shell of the dual-phase scaffold with randomly oriented core fibers boosted the initial adhesion and viability of cells. The created FSPCL/ESPCL-Gel 3:1 scaffold (FS=centrifugal force spinning/RJS, ES=wet electrospinning, Gel=gelatin) supported tenogenic differentiation, maintained high mechanical strength, and enhanced cell viability and orientation. The combination of two distinct fiber production methods produced scaffolds that were reliable and repeatable and could be used for tendon tissue engineering applications [144].

Comment #2: Regarding multilayered scaffolds. Recently, the electrospinning technique has also been combined with 3D printing technology, creating fibrous meshes layered with printed structures. Additionally, layer-by-layer scaffolds are also considered promising alternatives for reproducing the native characteristic of tendon/ligament tissues. The authors should provide some comments on the point.

Author’s reply:

The authors thank the reviewer’s suggestion, in this way,  more examples were included within the manuscript.

The following text was added to the manuscript:

Line 845:

Additionally, the 3D printing technology has been combined with the electrospinning technique to produce fibrous meshes layered with printed structures. Touré et al. directly electrospuned poly (caprolactone)/poly (glycerol sebacate) (PCL/PGS) nanofibres for tendon/ligament applications onto 3D printed PCL/PGS constructs coated with bioactive glasses. In vitro biocompatibility was found to be increased by the presence of bioactive glasses, and sufficient mechanical qualities, such as Young’s modulus, were also noted [141].

Line 852:

Layer-by-layer scaffolds, in which hydrogels are stacked between electrospun, knitted, or braided structures, are also thought to be promising alternatives for the same purpose. Zhao et al. created a multilayered system with a sheet of fibrin gel supplied with basic fibroblast growth factor (bFGF) and mesenchymal stem cells (MSCs) sandwiched between two poly(lactide-co-glycolide) (PLGA) knitted constructions, for tendon repair. The scaffold that was implanted in a rat model, specifically in a critical-size Achilles tendon defect, showed adequate biomechanical characteristics and allowed the MSCs that were added to it to express tenogenic markers, which supported tissue repair and regeneration [142]. A 3D multi-layered composite scaffold made of layers of hydrogel loaded with MSCs and coated with synthetic electrospun nanofibres was proposed by Rinoldi et al.. It was established that the PCL and polyamide 6 (PA6) electrospun matrix provided the mechanical qualities and supported the entire build. The hydrogel layers, however, imitated a microenvironment favorable for cell encapsulation and proliferation since they were made of gelatin methacryloyl (GelMA) and alginate. In a specially designed bioreactor, mounted constructs were cultivated while receiving mechanical and biochemical stimulation. To encourage the tenogenic differentiation of MSCs, the concentration of the BMP-12 addition was optimized. Results obtained in vitro demonstrated the stimuli's beneficial effects on tenogenic differentiation, alignment, and proliferation [128]. The findings indicated that the suggested structure may be employed to design functional tendons.

Comment #3: Regarding Engineering of tissue interfaces. The main difficulty in this field is the design of gradient scaffolds with zonal structure, architectures, compositions, and mechanical properties. However, this aspect is crucial to develop systems which can guarantee the physiological bio-functionality of the tissue. The authors should provide some comments on the three different tissue regions: tendon/ligament, fibrocartilage interface and bone as well as the design of the tendon–muscle interface.

Author’s reply:

In line with the opinion of the reviewer, the main difficulty in tendon and ligament tissue engineering is the creation of a single structure with gradient mineralized and nonmineralized sections, different collagen interlacing, and hard-to-soft mechanical qualities. In fact, it is an important topic that needs to be addressed in this document.

The following text was added to the manuscript:

Line 895:

The main difficulty in tendon and ligament tissue engineering is the creation of a single structure with gradient mineralized and nonmineralized sections, different collagen interlacing, and hard-to-soft mechanical qualities [1]. To be more precise, tendon/ligament, fibrocartilage interface, and bone are the three distinct tissue regions that should be constructed. Calejo et al., study aforementioned, developed gradient scaffolds to replicate tendon-to-bone interface using aligned microfibres produced by wetspinning. Two formulations were taken into account for this use: PCL/gelatin to resemble tendon tissues and PCL/gelatin with nano-to-microsized hydroxyapatite (HAp) particles to simulate bone. The braiding of PCL/gelatin with PCL/gelatin/HAp microfibres resulted in a 3D gradient structure with a continuous topographical and chemical gradient, Figure 14. Overall, the findings strongly supported the viability of adopting straightforward fibre processing methods, like wetspinning, to precisely regulate the topography and composition of fibres while tailoring cellular responses. The construction of 3D fibrous scaffolds for tendon-to-bone regeneration was also made possible by the integration of modern textile processes [58].

Comment #4: My concern is the lack of combination of different techniques in this review. The combination of different techniques is considered crucial to produce scaffolds which can recapitulate the native tissue properties and promote tissue regeneration. The multilayered systems, where each compartment can be independently produced and tailored, have emerged as the most effective alternatives for tendon and ligament TE.

Author’s reply:

The authors agree with the reviewer and decided to include more examples of combined techniques for the scaffolds production.

The following text was added to the manuscript:

Line 827:

Overall, the electrospun nanofibrous construct are also a possible option to create a scaffold for tendon and ligament regeneration. It offers a suitable environment for biosignaling, supports proper cell attachment and proliferation, and prevents the adhesion of surrounding tissues. However, poor cell infiltration and a shortage of cell biding sites are still thought to be those technologies' principal drawbacks [1].

Recently, multilayered systems appeared as the most promising candidates for tendon and ligament tissue engineering. By combining the characteristics of various source materials, morphologies, and production techniques, this method creates composite structures that may be the most advantageous in terms of biological response as well as mechanical and structural capabilities [1]. Lately, systems for tendon and ligament engineering that combine electrospun fibrous mats with woven/knitted structures in multilayered scaffolds have been investigated. Rashid et al. created a patch intended for tendon repair with two exterior layers made of electrospun polydioxanone (PDO) fibrous meshes and PDO woven structure, and an inner layer of aligned electrospun PCL fibres. A sheep model used for in vivo testing revealed blood vessel development and infiltration of cells, primarily tendon fibroblasts, onto the electrospun layer. There were no reports of an exaggerated inflammatory response or surrounding tissue adhesion [140].

Line 882:

In a study by Guner et al., a dual-phase fibrous scaffold was created with the goal of enhancing the construct's potential for healing by merging fibrous mats made by rotary jet spinning (RJS) and wet electrospinning (WES). Aligned PCL fibers (Shell) made by RJS and randomly oriented PCL or PCL/gelatin fibers (Core) made by WES systems were combined to create dual-phase scaffolds. The scaffolds replicated the remodeling and repair phases of tendon healing. Studies conducted in vitro revealed that the aligned PCL fiber shell of the dual-phase scaffold with randomly oriented core fibers boosted the initial adhesion and viability of cells. The created FSPCL/ESPCL-Gel 3:1 scaffold (FS=centrifugal force spinning/RJS, ES=wet electrospinning, Gel=gelatin) supported tenogenic differentiation, maintained high mechanical strength, and enhanced cell viability and orientation. The combination of two distinct fiber production methods produced scaffolds that were reliable and repeatable and could be used for tendon tissue engineering applications [144].

Line 917:

All in all, combining approaches to create scaffolds with higher structural and biomechanical qualities is fairly frequent. With this approach, the scaffold reflects a higher degree of complexity, that will more closely resemble a natural tendon or ligament [2].

Reviewer 2 Report

The manuscript submitted by Rocha et. al. aims to discuss recent polymeric nanofibers produced by wet-spinning in order to mimic, repair, and replace the tendons and ligaments. The first four chapters are general, presenting the main characteristics of the structure, composition, and treatment strategies for tendons and ligaments, the manufacturing technique (wet-spinning), and general aspects regarding biodegradable polymers. Most of the information is already known and presents low interest for the reader, but it is well-written and easy to follow.

The most interesting part starts in chapter 5, where the authors present the three most important biodegradable polymers used to produce continuous fibers by wet-spinning (PHAs, PCL, and PEG). However, in this part, some examples where electrospinning was used were also presented, which might be confusing and in disagreement with the subsection’s title. I suggest removing these parts or changing the title to „spinning”, not „wet-spinning”.

Author Response

Dear Editor and Dear Reviewers,

Please find enclosed the revised version of the above referenced manuscript. The authors wish to thank all the reviewers for their valuable comments and inputs, having reviewed and edited the original manuscript accordingly, while highlighting all major changes (in yellow).

This document also contains a detailed, point to point answer to all the reviewers’ remarks and indications, after this cover letter.

The authors also had into consideration the Editor comments about some paragraphs containing similarities with other works, in that sense several paragraphs were rewritten and corrected as requested.

Looking forward to hearing from you,

 Yours sincerely,

Doctor Diana P. Ferreira

Researcher at Minho University

All reviewers’ comments have been acknowledged and analysed. This document contains a point-to-point answer to each comment, while mentioning any relevant edits to the manuscript.

Reviewer #2:

Comment #1: The manuscript submitted by Rocha et. al. aims to discuss recent polymeric nanofibers produced by wet-spinning in order to mimic, repair, and replace the tendons and ligaments. The first four chapters are general, presenting the main characteristics of the structure, composition, and treatment strategies for tendons and ligaments, the manufacturing technique (wet-spinning), and general aspects regarding biodegradable polymers. Most of the information is already known and presents low interest for the reader, but it is well-written and easy to follow.

The most interesting part starts in chapter 5, where the authors present the three most important biodegradable polymers used to produce continuous fibers by wet-spinning (PHAs, PCL, and PEG). However, in this part, some examples where electrospinning was used were also presented, which might be confusing and in disagreement with the subsection’s title. I suggest removing these parts or changing the title to „spinning”, not „wet-spinning”.

Author’s reply:

The authors are very thankful for the reviewer comments. In fact, some information is already known in this manuscript because a review article it is supposed to cover all the most important works that fall within the scope of the review. As long as the authors know, there is no evidence in literature of a review covering all the sections described in this manuscript. However, in order to increase the interest of the readers and taking into consideration the reviewer opinion about the “confusing and disagreement with the subsection title”, the authors decided to add a new section to enrich and clarify the manuscript entitled “7. Spinning”, which includes the examples of the use of electrospinning technique for tendon and ligament TE

The following text was added to the manuscript:

Line 738:

It is known that fibrous scaffolds created with different fibre-based fabrication techniques are suitable for replacing anisotropic tissues and accelerating their healing. Wet and electrospun constructs are the most often used fibre-based scaffolding technologies for tendon and ligament tissue regeneration [1]. In recent years, electrospinning techniques have also been used to process biodegradable polymers, such as PHAs, intended for biomedical applications [134]. The relationship between processing parameters and electrospun fibre assembly architecture was examined in one of the earliest publications on electrospinning of PHA, which was published in 2006 [135]. The investigation led to the creation of a number of scaffolds with an average fibre diameter of a few microns, made of PHB, PHBV, or their combination (75:25, 50:50, or 25:75 weight ratio). A phase inversion mechanism associated with the swift evaporation of the solvent used, chloroform, was proposed by the authors as the explanation for the rough surface that SEM analysis of PHB/PHBV blend fibres revealed. These scaffolds sustained the in vitro growth of mouse fibroblasts (L929) and human osteoblasts (SaOS-2), at higher levels than analogous cast-films, indicating biocompatibility of these materials to both types of cells. As said, the viability of the cells cultured with an extraction medium is reported in Figure 11 in terms of the relative absorbance with respect to the absorbance value of SaOS-2 that were cultured with fresh SFM for the same culture period [136].

Line 762:

Chen et al., fabricated a bioactive cartilage tissue engineering scaffold that promotes cartilage regeneration. In this study, electrospun PHBV fibrous scaffolds were modified with QUE to enhance the bioactivity of PHBV. The modified PHBV fibrous scaffolds promoted the proliferation of chondrocytes, maintained the chondrocytic phenotype, and facilitated the formation of cartilage ECM. More importantly, the PHBV-g-QUE fibrous scaffold significantly promoted maturation of neo-cartilage tissue and cartilage regeneration in vivo compared with the neat PHBV fibrous scaffold (Figure 12). The results suggested that the PHBV-g-QUE fibrous scaffold can potentially be applied in cartilage tissue engineering (CTE) [137].

Line 778:

Wu et al., successfully developed tailored nanofibrous scaffolds, by the electrospinning process, of hydroxyapatite (HAp) dispersed in a polycaprolactone/chitosan (HAp-PCL/CHT) nanofibrous matrix for tendon and ligament TE. Favorable mechanical properties (load and modulus), cellular responses and biocompatibility were achieved. The load and modulus of the produced HAp-PCL/CHT fibres was 250.1 N and 215.5 MPa, which is very similar to the standard value of the human tendon and ligament tissues. The cellular responses and biocompatibility of the nanofibrous scaffolds were investigated with human osteoblast (HOS) cells and the microscopic images clearly showed that the HOS cells were well attached and flatted on the scaffolds (Figure 13 (a), (a1) and (b)). It was also exhibited that the HAp dispersed PCL/CHT nanofibrous scaffolds promoted higher adhesion and proliferation of HOS cells comparable to the nanofibrous scaffolds without HAp nanoparticles (Figure 13 (c)). Overall, the physical and biological properties of the synthesized HAp-PCL/CHT scaffolds were very close to that of the normal human tendon and ligament [87].

Line 799:

More, Domingues et al. enhanced the biomechanical performance of anisotropically aligned electrospun nanofibrous scaffolds based on poly-3-caprolactone/chitosan (PCL/CHT) by incorporating small amounts of CNC (up to 3 wt%). The aligned PCL/CHT/CNC nanocomposite fibrous scaffolds met not only the mechanical requirements for tendon TE applications but also provided tendon mimetic extracellular matrix (ECM) topographic cues, which is a key feature for maintaining tendon cell’s morphology and behavior [28].

Line 806:

In a study made by Toncheva et al., two types of antibacterial micro- and nanofibrous mats based on PLA and PEG were prepared by electrospinning. PEG incorporation was achieved by its physical blending with or chemical grafting on PLA. Physical blending or chemical grafting showed a plasticizing effect on PLA but did not significantly modify the wettability of the mats. The PEG incorporation method also influenced the mechanical properties of the mats (Table 7). Fibrous mats of physically blended PLA and PEG proved quasi-ductile behavior, while brittle behavior was registered for the chemically grafted mats [138].

Line 816:

On another study by Huang et al., PCL/PEO nanofibres with or without Dipsacus asper Wall extracts (DAE) were produced under optimal electrospinning conditions with the aim to evaluate the osteogenic differentiation of periodontal ligament stem cells (PDLSCs) of DAE. The physical property analysis of the obtained nanofibres included Fourier transform infrared spectroscopy (FTIR), mechanical strength, biodegradability, swelling ratio and porosity, and cell compatibility. The results of the study confirmed that both DAE and PCL/PEO nanofibres have the effect of promoting osteogenic differentiation. However, the addition of DAE effectively increased the elongation, swelling, porosity, and degradability of the PCL/PEO nanofibres, resulting in fibres with better applicability as tissue engineering scaffolds and increased osteogenesis induction effects [139].

Round 2

Reviewer 1 Report

Thank you for providing answers to my questions and doing the corrective actions accordingly.

I appreciate the effort of the authors.

I have read in detail the responses and they address the issues raised.

It appears much stronger and suitable for publication.

Author Response

The authors are very grateful for the reviewers' comments.